# Learning From Biased Soft Labels

**Hua Yuan**[1,2], **Yu Shi**[1,2], **Ning Xu**[1,2], **Xu Yang**[1,2], **Xin Geng**[1,2*], **Yong Rui**[3*]

[1] School of Computer Science and Engineering, Southeast University, Nanjing 210096, China
[2] Key Laboratory of New Generation Artificial Intelligence Technology and Its
Interdisciplinary Applications (Southeast University), Ministry of Education, China
[3] Lenovo Research, Beijing 100085, China
{yuanhua,seushiyu,xning,xuyang_palm,xgeng}@seu.edu.cn, yongrui@lenovo.com

## Abstract

Since the advent of knowledge distillation, many researchers have been intrigued by the *dark knowledge* hidden in the soft labels generated by the teacher model. This prompts us to scrutinize the circumstances under which these soft labels are effective. Predominant existing theories implicitly require that the soft labels are close to the ground-truth labels. In this paper, however, we investigate whether biased soft labels are still effective. Here, bias refers to the discrepancy between the soft labels and the ground-truth labels. We present two indicators to measure the effectiveness of the soft labels. Based on the two indicators, we propose moderate conditions to ensure that, the biased soft label learning problem is both *classifier-consistent* and *Empirical Risk Minimization* (ERM) *learnable*, which can be applicable even for large-biased soft labels. We further design a heuristic method to train Skillful but Bad Teachers (SBTs), and these teachers with accuracy less than 30% can teach students to achieve 90% accuracy on CIFAR-10, which is comparable to models trained on the original data. The proposed indicators adequately measure the effectiveness of the soft labels generated in this process. Moreover, our theoretical framework can be adapted to elucidate the effectiveness of soft labels in three weakly-supervised learning paradigms, namely incomplete supervision, partial label learning and learning with additive noise. Experimental results demonstrate that our indicators can measure the effectiveness of biased soft labels generated by teachers or in these weakly-supervised learning paradigms.

## 1   Introduction

Knowledge distillation [2, 17, 20] has achieved remarkable achievements in a wide range of applications. It has emerged as a popular paradigm for model compression [22, 21] and transfer learning [41, 34] by distilling knowledge from the big model (teacher) to the small model (student). Students inherit the knowledge of the teacher by imitating the soft labels generated by the teacher model. Many experiments show that learning from soft labels can be very effective, even surpassing learning from ground-truth labels [2, 17, 37]. However, there are still many mysteries why these soft labels are effective. Most existing theories do not apply to soft labels that significantly deviate from the ground-truth labels and lack explicit indicators to evaluate the soft labels [39, 58, 7, 35]. In this paper, we mainly focus on the effectiveness of the biased soft labels, especially large-biased soft labels.

It is significant to study the effectiveness of biased soft labels. Firstly, expert-labeled annotation typically requires a substantial investment of manpower and time. But the soft labels in weakly supervised learning could be cheap and biased [59, 46]. Evaluating the biased soft labels is instructive for weakly supervised learning. Secondly, these biased soft labels can apply for privacy protection

---

*Corresponding author

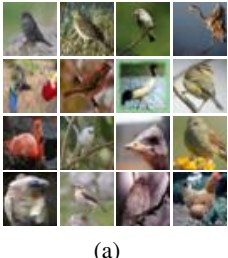 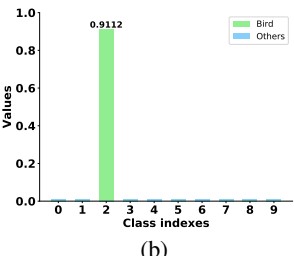 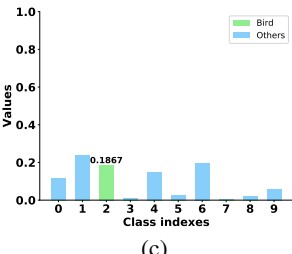

|(a)|(b)|(c)|

Figure 1: (a) Images of the birds in CIFAR-10. (b) An example of the soft label for "Bird" in Yuan et al. [53]. (c) An example of the large-biased soft label for "Bird". Compared with Yuan et al. [53], the value of the ground-truth label (Bird) in the large-biased soft label is probably much smaller, and is likely not the top of the large-biased soft label.

[15, 11]. When they are employed for subsequent training, they can conceal the ground-truth labels. It is consistent with the requirements of privacy and utility in privacy protection [27].

After realizing the significance of biased soft labels, we study the biased soft labels and reveal that large-biased soft labels can also teach good students. This phenomenon inspire us to seek indicators to measure the effectiveness of soft labels. As a result, we propose two intuitive indicators, namely *unreliability degree* and *ambiguity degree*. The intuition behind them is to convert the soft label into a top-$k$ set ($k$ is a constant), which contains the top-$k$ labels of the soft label. Unreliability degree is the probability that the ground-truth label is not in the top-$k$ set while ambiguity degree is the upper bound of the probability that the incorrect label and the correct label co-occur in the top-$k$ set.

Furthermore, based on the proposed indicators, we present moderate conditions to ensure that, the biased soft label learning problem is classifier-consistent [28] and Empirical Risk Minimization (ERM) learnable [6]. The former guarantees that the learners can converge to the optimal solution as learning from original labeled data. The latter implies that learners' performance can generalize to the entire data distribution. Our theory not only guarantees the effectiveness of biased soft labels, but also provides explicit indicators to evaluate the soft labels.

Biased soft labels are prevalent in weakly supervised learning [24, 44, 59], and our theory can offer theoretical insights for these paradigms. In detail, we apply the theory to three classic weakly-supervised learning paradigms: incomplete supervision [3], partial label learning [5] and learning with additive noise [10]. In these weakly-supervised learning paradigms, soft labels are biased and we provide a theoretical guarantee for the learners in these fields. In other words, we prove that the biased soft labels in the three paradigms are effective to train a good model.

It is important to note that, Yuan et al. [53] argues that poorly-trained teachers can teach good students. In fact, their soft labels are a mixture of the teacher's outputs and the ground-truth labels with a ratio of $0.1 : 0.9$, which makes the soft labels close to the ground-truth labels as illustrated in Figure 1(b). Differently, our experiments reveal that large-biased soft labels can also teach good students. Furthermore, the effectiveness of these soft labels can be measured by unreliability degree and ambiguity degree. In addition, experiments on three weakly-supervised learning paradigms also demonstrate that biased soft label learning problem is learnable and the proposed indicators are effective. Our contributions can be summarized as follows:

- We find that learning from large-biased soft labels may also achieve comparable performance and intend to explore the underlying mechanisms behind the effectiveness of the biased soft labels.
- Two indicators have been proposed to measure the effectiveness of soft labels. Based on the indicators, we present moderate conditions to guarantee the effectiveness of the soft labels. It is proved that the biased soft label learning problem is classifier-consistent and ERM learnable.
- A heuristic method is designed to train skillful but bad teachers, i.e., teachers with low accuracy but who can teach good students. We can explain this phenomenon with unreliability degree and ambiguity degree.
- The theory provides a theoretical view for the learners in three weakly-supervised learning paradigms. Specifically, we provide theoretical guarantees for the learnability of these paradigms from the perspective of soft labels. Experimental results are consistent with our theory.

## 2 Related Work

**Knowledge Distillation and Label Smoothing** Knowledge Distillation (KD) was initially proposed in model compression [22, 21] and then applied to transfer learning [41, 34]. There is growing interest in why distilling can transfer information and what the *dark knowledge* hidden in the soft labels is. Furthermore, the dark knowledge also exists in the variants of knowledge distillation, which are introduced in the appendix A.1. Label Smoothing (LS) [38] is a regularization method to improve performance by mixing the uniform noise into the ground-truth label. It is convinced that soft labels can restrain overconfidence of the student model. Essentially, both KD and LS can be unified as learning from soft labels Yuan et al. [53].

**Label Enhancement** Label Distribution Learning (LDL) [13] was proposed to exploit the label distribution to mirror the relationship between the label and the instance, where the formalization of the label distribution is identical to the soft labels mentioned above. In the remainder of the paper, we use nomenclature soft labels. Due to the high cost of labeling the soft labels, Label Enhancement (LE) [48] was proposed to recover the soft label from the logical label by exploiting the implicit correlation among different labels. Numerous novel algorithms have been designed in recent years that aim to improve the predictive model with the soft labels [47, 57]. Wang and Geng [42] applied the margin theory to the soft labels and designed the adaptive margin loss.

In fact, most existing explanations of the soft labels are empirically and experimentally validated, while the rigorous theoretical analyses usually have strong assumptions regarding the model or data distribution. Phuong and Lampert [35] explored the mechanism of distillation where the teacher model and the student model are linear. Allen-Zhu and Li [1] supposed that the instance could be decomposed into multiple independent features and had a linear relationship with the sample, and then proved the effectiveness of the soft labels. Wang and Yoon [43] solved the objective functional problem of self-distillation with the Green's function, which assumes that the network can reach the optimal solution. Menon et al. [31] and Zhou and Song [58] regarded the generated soft labels as the posterior probability and assumed the existence of the Bayes probability. There is also some work analyzing soft labels from the perspective of transfer risk [19, 18]. Most existing theories suggest that students can achieve good performance only when the soft labels are close to the ground-truth labels, and lack explicit indicators to measure the effectiveness of the soft labels.

## 3 Methodology

### 3.1 Preliminary

Let $\mathcal{X}$ be the instance space, $\mathcal{Y} = \{1, 2, \ldots, c\}$ be the label space with $c$ classes and $\Delta$ be the soft-label space over $\mathcal{Y}$, i.e., $\Delta = \{\boldsymbol{d} \in \mathbb{R}^c | \sum_{i \in \mathcal{Y}} \boldsymbol{d}_i = 1, \boldsymbol{d}_i \geq 0 \text{ for all } i \in \mathcal{Y}\}$. Define $\mathcal{D}$ as the data distribution over $\mathcal{X} \times \mathcal{Y}$ and $\mathcal{H}$ as the hypothesis space from $\mathcal{X}$ to $\mathcal{Y}$. Each $h \in \mathcal{H}$ is called the learner or hypothesis. When $h \in \mathcal{H}$ is a neural network, it often employs a softmax function in the final layer and takes the highest value as the prediction. Slightly abusing notation, we use $h(\boldsymbol{x}) \in \Delta$ to denote the soft label generated by model $h$, and $\tilde{h}(\boldsymbol{x}) \in \mathcal{Y}$ to represent the model's prediction.

Next, we introduce some important concepts in the Probably Approximately Correct (PAC) learning [40]. The Natarajan dimension [32] was proposed to characterize the capacity of multiclass hypothesis spaces and we denote $d_{\mathcal{H}}$ as the Natarajan dimension of $\mathcal{H}$. The expected classification error of a hypothesis $h \in \mathcal{H}$ is defined as $\text{Err}_{\mathcal{D}}(h) = \mathbb{E}_{(\boldsymbol{x}, y) \sim \mathcal{D}}[\mathbb{I}(h(\boldsymbol{x}) \neq y)]$, where $\mathbb{I}(\cdot)$ is the indicator function. Given a finite dataset $\mathbf{z}$, the empirical classification error is defined as $\text{Err}_{\mathbf{z}}(h) = \frac{1}{n} \sum_{i=1}^{n} \mathbb{I}(h(\boldsymbol{x}) \neq y)$. The Empirical Risk Minimization (ERM) learner $h^{\text{ERM}}$ is defined as $h^{\text{ERM}} = \underset{h \in \mathcal{H}}{\arg\min} \, \text{Err}_{\mathbf{z}}^{\Delta}(h|f)$, while the optimal learner $h^*$ is defined as $h^* = \underset{h \in \mathcal{H}}{\arg\min} \, \text{Err}_{\mathcal{D}}(h)$.

Considering that this study involves two models: generating soft labels and learning from the soft labels. To avoid ambiguity, we use $f$ to represent the model generating soft labels (teacher) and $h$ to denote the model trained with soft labels (student). It should be noted that $f \in \mathcal{H}$ can be a neural network or a map based on certain rules, such as label smoothing.

In order to gain a deeper insight into the concept of biased soft labels, we have provided precise definitions for relevant terms below. These definitions may not have a direct correlation with the theoretical analysis presented in section 3.2, but they serve to elucidate the boundaries of our research.

**Definition 1** (Bias of soft labels). *Give a dataset $\mathcal{D}$ consisting of $n$ samples, the feature vector for the $i$-th sample is denoted as $\boldsymbol{x}_i$ and the corresponding label is denoted as $y_i$. Let $f$ represent a model or a mapping rule. The bias of the soft labels generated by $f$ on dataset $\mathcal{D}$ is*

$$\mathrm{Bias}(f, \mathcal{D}) = \frac{1}{n} \sum_{i=1}^{n} [1 - f_{y_i}(\boldsymbol{x}_i)],$$

*where $f_{y_i}(\boldsymbol{x}_i)$ refers to the component of the soft label $f(\boldsymbol{x}_i)$ that corresponds to the true label $y_i$.*

**Definition 2** (Large-biased soft labels). *Soft labels generated by $f$ on dataset $\mathcal{D}$ is called biased soft labels when $\mathrm{Bias}(f, \mathcal{D}) > 0$ and called large-biased soft labels when $\mathrm{Bias}(f, \mathcal{D}) \geq 1$.*

The bias, as defined here, essentially represents the disparity between the soft label and the true label. Here, we employ the Manhattan distance, also referred to as L1-norm, although other metrics can also be suitable. A threshold of $\mathrm{Bias} \geq 1$ is employed to categorize soft labels as "large-biased." In practical scenarios, the choice of the threshold can be adapted based on specific contexts. After establishing the concept of large-biased soft labels, we proceed to define "bad teachers."

**Definition 3** (Bad teachers). *We define $f$ as a bad teacher if the soft labels it generates on dataset $\mathcal{D}$ are large-biased. Typically, $\mathcal{D}$ is the training set for $f$.*

The definition of the bad teachers is based on the definition of large-biased soft labels. Similarly, here we adapt 1 as the boundary value for bad teachers. It's important to note that when we use "good/bad" to describe teachers, we are referring to the performance of teachers on the dataset $\mathcal{D}$. On the other hand, when we say teachers are "skillful," we are emphasizing that they are adept at instructing students, leading to excellent student performance. Due to the fact that the performance of the student model is contingent on both the model structure and the complexity of the dataset, we are unable to establish precise boundaries for "skillful teachers." In this context, 'skillful' merely signifies that the teacher produces students with acceptable performance.

## 3.2 Theoretical Analysis of Soft Labels

In this subsection, we define two indicators, unreliability degree and ambiguity degree, to measure the effectiveness of soft labels. Based on these indicators, we present moderate conditions to ensure that the biased soft label learning problem is classifier-consistent and Empirical Risk Minimization (ERM) learnable. The underlying intuition behind the indicators is to convert the soft label into a set. Let $\Omega_k(\boldsymbol{d}) = \{i \in \mathcal{Y} \mid i \text{ ranks top-}k \text{ in } \boldsymbol{d}\}$ be the set of top $k$ labels in the soft label $\boldsymbol{d}$. Here, $k$ is a constant ranging in $\{1, 2, \ldots, c-1\}$. When $k = 1$, $\Omega_k(\boldsymbol{d})$ has only one element, i.e., the prediction. For soft labels generated by teacher $f$, we define the *unreliability degree* as,

$$\eta_k(f) = \mathrm{Pr}_{(\boldsymbol{x},y) \sim \mathcal{X} \times \mathcal{Y}}(y \notin \Omega_k(f(\boldsymbol{x}))). \tag{1}$$

However, it is not enough to measure the soft labels by merely unreliability degree. For example, for images whose ground-truth labels are 1, if label 2 always appears in $\Omega_k(\boldsymbol{d})$, then the student model is unable to distinguish label 1 from label 2. Therefore, we introduce the *ambiguity degree* [4] and extend it to more general soft labels (induced by teacher $f$)

$$\gamma_k(f) = \max_{i \in \mathcal{Y}} \mathrm{Pr}_{(\boldsymbol{x},y) \sim \mathcal{X} \times \mathcal{Y}, y \neq i}(i \in \Omega_k(f(\boldsymbol{x}))). \tag{2}$$

Ambiguity degree bound the probability of *co-occurrence*. In other words, if a model $f$ is with ambiguity degree $\gamma_k(f)$, then $\mathrm{Pr}(i \in \Omega_k(f(\boldsymbol{x})) \mid i \neq y, x, y) \leq \gamma_k(f)$. The smaller $\eta$ or $\gamma$ is, the more effective the soft labels are. However, when $k$ increases, $\eta$ will decrease and $\gamma$ is inverse, which means $k$ should be selected cautiously.

**Theorem 1.** *Training with soft labels generated by the teacher model $f$, if $\gamma_k(f) < 1 - \frac{\eta_k(f)}{1 - \eta_k(f)}$, then the optimal student model $h^* \in \mathcal{H}$ satisfies $h^* = \underset{h \in \mathcal{H}}{\arg\min}\, \mathrm{Err}_{\mathcal{D}}(h|f)$.*

The proof can be found in A.2. Theorem 1 ensures that the student model $h$ learning from the teacher can converge to the optimal learner over the entire data distribution. This property is known as classifier-consistency [12]. However, it does not provide the sample complexity bounds of the learning problem. In other words, it does not establish a connection between the generalization bound and the number of training samples.

Next, we provide our main result, a sufficient condition for the ERM learnability of the biased soft label learning problem. In the previous subsection, we presented the definition of expected classification error and empirical classification error, which are based on the label space $\mathcal{Y}$. Here, given the soft label generating model $f$, we define the soft label-based expected error

$$\mathrm{Err}^{\Delta}(h|f) = \mathbb{E}_{(\boldsymbol{x},y)\sim\mathcal{X}\times\mathcal{Y}}[\mathbb{I}(\tilde{h}(\boldsymbol{x}) \notin \Omega_k(f(\boldsymbol{x})))],$$

and the soft label-based empirical error on dataset $\mathbf{z} = \{(\boldsymbol{x}_i, y_i)\}_{i=1}^n$

$$\mathrm{Err}_{\mathbf{z}}^{\Delta}(h|f) = \frac{1}{n}\sum_{i=1}^{n}\mathbb{I}\left(\tilde{h}\left(\boldsymbol{x}_i\right) \notin \Omega_k(f(\boldsymbol{x}_i))\right).$$

In the above equations, the set $\Omega_k(f(\boldsymbol{x}_i))$ is determined by the soft label $f(\boldsymbol{x}_i)$. We denote $H_\eta$ as the set of teacher models whose generated soft labels are with unreliability degree $\eta$, i.e., $H_\eta = \{f : \eta_k(f) = \eta\}$. With such soft labels, we analyze the performance of the ERM learners (students). The definition of the ERM learner is provided in the preliminary. Specifically, based on the unreliability degree in (1) and the ambiguity degree in (2), we provide a sufficient condition to ensure that, the biased soft label learning problem is ERM learnable.

**Theorem 2** (Main theory). *For $k \in \{1, 2, \ldots, c-1\}$, assume the unreliability degree $\eta_k(f)$ and the ambiguity degree $\gamma_k(f)$ of the soft labels generated by teacher model $f$, denoted concisely as $\eta_k$ and $\gamma_k$, and satisfy $0 < \eta_k, \gamma_k < 1$ and $\eta_k + \gamma_k < 1$. Let $\theta_k = \log\frac{2(1-\eta_k)}{1-\eta_k+\gamma_k}$ and suppose the Natarajan dimension of the hypothesis space $\mathcal{H}$ is $d_{\mathcal{H}}$. Define*

$$n_0(\mathcal{H}, \varepsilon, \delta) = \min_{k\in\{1,2,\ldots,c-1\}}\left\{\frac{2}{\frac{\theta_k\varepsilon}{2} + \log\frac{1}{2-2\eta_k}}(d_{\mathcal{H}}(\log(2d_{\mathcal{H}}) \right.$$
$$\left. + \log\frac{1}{\frac{\theta_k\varepsilon}{2} + \log\frac{1}{2-2\eta_k}} + 2\log L) + \log\frac{1}{\delta} + 1)\right\}.$$

*Then when $n > n_0$, the ERM learner satisfies $\mathrm{Err}_{\mathcal{D}}(h^{ERM}|f) < \varepsilon$ with probability $1 - \delta$.*

The proof can be divided into two lemmas. Let define $H_\varepsilon$ be the set of hypotheses with error at least $\varepsilon$, i.e, $H_\varepsilon = \{h \in \mathcal{H} : \mathrm{Err}_{\mathcal{D}}(h|f) \geq \varepsilon\}$. Our target is to bound the probability $\Pr(h \in H_\varepsilon)$, which measures the generalization of the learner $h$. Since the entire soft label space is inaccessible, $H_\varepsilon$ is evaluated by the mediator $R_{n,\varepsilon}$ as follows:

$$R_{n,\varepsilon} = \left\{\mathbf{z} \in (\mathcal{X} \times \mathcal{Y})^n : \exists h \in H_\varepsilon, \mathrm{Err}_{\mathbf{z}}^{\Delta}(h|f) = 0\right\}.$$

Then, our goal is to prove that $\Pr(\mathbf{z} \in R_{n,\varepsilon} \mid f \in H_\eta) \leq \delta$. In other words, given dataset $\mathbf{z}$, the student model $h$ has the generalization bound greater than $\varepsilon$ with probability at most $\delta$. Since the teacher model $f$ is intractable and the soft label space $\Delta$ is unknown, it is very difficult to directly calculate the conditional probability $\Pr(\mathbf{z} \in R_{n,\varepsilon} \mid f \in H_\eta)$. We bound it by introducing a testing set $\mathbf{z}'$. Lemma 1 is adapted from [[9], Lemma 11.1.5] and [[16], Corollary 2.6].

**Lemma 1.** *For a testing set $\mathbf{z}' \in (\mathcal{X} \times \mathcal{Y})^n$, we can define the set $S_{n,\varepsilon}$ as*

$$S_{n,\varepsilon} = \left\{(\mathbf{z}, \mathbf{z}') \in (\mathcal{X} \times \mathcal{Y})^{2n} : \exists h \in H_\varepsilon, \mathrm{Err}_{\mathbf{z}}^{\Delta}(h|f) = 0, \mathrm{Err}_{\mathbf{z}'}^{\Delta}(h|f) \geq \frac{\varepsilon}{2}\right\}.$$

*Then, we have $\Pr((\mathbf{z}, \mathbf{z}') \in S_{n,\varepsilon} \mid f \in H_\eta) \geq \frac{1}{2}\Pr(\mathbf{z} \in R_{n,\varepsilon} \mid f \in H_\eta)$, for $n > \frac{8\log 2}{\varepsilon}$.*

The proof of Lemma 1 can be found in appendix A.3. With lemma 1, we just need to estimate $\Pr((\mathbf{z}, \mathbf{z}') \in S_{n,\varepsilon} \mid f \in H_\eta)$. It seems more complicated to introduce the testing set $\mathbf{z}'$ but we can swap training/testing instance pairs, which is a classic method in the proof of learnability, to refine the data distribution on $\mathcal{X} \times \mathcal{Y}$ into each single instance.

**Lemma 2.** *On the same condition of theorem 2, we have*

$$\Pr\left((\mathbf{z}, \mathbf{z}') \in S_{n,\varepsilon} \mid f \in H_\eta\right) \leq (2n)^{d_{\mathcal{H}}} L^{2d_{\mathcal{H}}} \exp\left(-\frac{n\theta\varepsilon}{2}\right).$$

The proof of Lemma 2 can be found in appendix A.4. With Lemma 1 and Lemma 2, we can prove Theorem 2, which is detailed in the appendix A.5.

In this section, we establish two essential properties of the soft labels. Classifier-consistency guarantees the effectiveness of the soft labels in a macroscopic perspective, and ERM learnability provides a microcosmic generalization bound for the student model $h$. The corresponding threshold conditions are presented to ensure the student model can learn from the soft labels. The theory is applicable to all biased soft labels. This inspires us to consider that accuracy alone may not be a sufficient measure of a teacher's teaching ability. In the section 5.1, we design a simple heuristic algorithm to generate such skillful but bad teachers, i.e., teachers with low accuracy but can teach good students. These results are illustrative for the comprehension and development of the soft label based algorithms.

## 4 Biased Soft Labels in Weakly-Supervised Learning

The labels in weakly-supervised learning (WSL) could be incomplete, inexact, inaccurate [59] because accurate annotation is often expensive and difficult to obtain. Soft labels are widely used in WSL. Many weakly-supervised learning paradigms can be transformed into learning from soft labels. Due to the lack of the supervisory information, the soft labels could be large-biased but the model can still learn from them. In this section, our theory is adapted to three classic weakly-supervised learning paradigms and can provide theoretical guarantee for the learnability of these problems. These findings reflect that the theory is promising and extensible.

### 4.1 Incomplete Supervision

In incomplete supervision, there are labeled data and unlabeled data. A common approach is to use the model to label the unlabeled data and then learn with all data iteratively. The soft labels of the unlabeled data are probably biased but make a significant contribution to the training process. We propose an ideal accuracy function and, based on it, we provide a theoretical analysis on incomplete supervision from the perspective of soft labels. Suppose there are $N$ labeled data and $M$ unlabeled data sampled from $\mathcal{X} \times \mathcal{Y}$. The predictive model $h$ is an ERM learner on both labeled data and unlabeled data. The label of unlabeled data will be updated iteratively.

**Assumption 1.** *For $N$ labeled data and $M$ unlabeled data whose soft labels have unreliability degree $\eta$ and ambiguity degree $\gamma$, the ERM learner $h$ has a deterministic accuracy funtion $\rho(\eta, \gamma)$, the probability that $h$ predict correctly. The model architecture, data distribution and optimization are implicitly included in $\rho(\eta, \gamma)$.*

**Assumption 2.** *Given that smaller values of $\eta$ or $\gamma$ result in more supervised information in the soft labels, we assume $\rho(\eta, \gamma)$ decreases with $\eta$ and $\gamma$.*

We suppose that the incorrect labels share the equal probability to appear in the top-$k$ set. The distribution of the incorrect labels can be characterized in a more refined manner. For instance, assume that there is a upper bound of $\frac{p(i|\boldsymbol{x})}{p(j|\boldsymbol{x})}, i, j \in \mathcal{Y}, i, j \neq y, i \neq j$. We employ simplified assumptions because they can still capture and reflect this process. Based on the ideal $\rho(\eta, \gamma)$, we delineate the progressive performance of $h$. Let $\rho_t$ denote the accuracy of $h$ at epoch $t$ and we have

$$\rho_{t+1} \geq \rho(1 - \rho_t, \frac{c - k - \rho_t}{c - 1}). \tag{3}$$

In practice, as learner $h$ learns from labeled data and unlabeled data, the performance of $h$ will improve and the soft labels of the unlabeled data will be more effective. Consequently, $h$ and the soft labels may achieve a dynamic equilibrium.

**Theorem 3.** *Based on the ideal accuracy function $\rho(\eta, \gamma)$, with a moderate initial state $\eta_0, \gamma_0$ satisfying Theorem 2, if final accuracy of $\rho_{final}$ exists, it can be calculated by the following fixed point equation:*

$$x = \rho(1 - x, \frac{c - k - x}{c - 1}). \tag{4}$$

*where $k$ accords with the top-$k$ set in $\eta$ and $\gamma$, $c$ is the number of class labels. If $\rho(\eta, \gamma)$ is $k_L$-Lipschitz continuous ($k_L < 1 - \frac{1}{c}$), then $\rho_{final}$ exists and is unique.*

The proof is detailed in A.6. In fact, the deterministic $\rho(\eta, \gamma)$ is unattainable due to the indeterminacy of the optimization and the potential uncertainty in the data distribution. An intuitive extension is to assume $\rho(\eta, \gamma)$ is a probability distribution related to the training specifics, which can be

further investigated. Theorem 3 is coarse yet in agreement with the general intuition. The model $h$ improves as the soft labels envolve and finally reach the bottleneck restricted by the model, data and optimization.

In this section, we demonstrate the potential benefits of our theory in several classic weakly-supervised learning paradigms. There remain numerous domains associated with soft labels. It is essential to possess an appropriate theory to analyze the soft labels for the corresponding algorithms. Our theory can be instrumental for comprehending and constructing the soft label based algorithms.

## 4.2 Partial Label Learning

In Partial Label Learning (PLL), each instace is typically assigned a set of possible labels, i.e., the candidate label set $s$ [28, 46, 49]. The corresponding candidate label space is denoted by $\mathcal{S}$, i.e., the non-empty power set of $\mathcal{Y}$. Traditional PLL assumes the ground-truth label must be in the candidate label set. But recently, Lv et al. [29] considers that ground-truth label could be not in the candidate label set, which is named as Unreliable Partial Label Learning (UPLL).

In UPLL, there are two basic concepts, partial rate $\nu$ and unreliable rate $\mu$. Partial rate $\nu$ is the ratio of incorrect labels in the candidate label set to total labels. A lower partial rate usually indicates a better performance of the model. Unreliable rate $\mu$ is the probability of ground-truth label $y$ not in the candidate label set, which can be formally stated as

$$\mu = \Pr_{(\boldsymbol{x}, y, s) \sim \mathcal{X} \times \mathcal{Y} \times \mathcal{S}} (y \notin s).$$

The discrete candidate label set $s$ can be transformed into the soft label by

$$d_i = \begin{cases} \dfrac{1}{|s|} & i \in s \\ 0 & i \notin s \end{cases},$$

where $|s|$ is the cardinality of set $s$. So PLL also can be viewed as learning from biased soft labels. Then we have the following corollary.

**Corollary 1.** *For UPLL with partial rate $\nu$ and unreliable rate $\mu$, regard the generated soft labels as teacher. Then, we have $\eta = \mu$ and $\gamma = \nu$. With the same conditions in Theorem 2, UPLL is ERM learnable and the sample complexity remains unchanged.*

This corollary show that UPLL is ERM learnable under a moderate condition. We provide new insights from the perspective of soft labels.

## 4.3 Learning with Additive Noise

Additive noise mechanism [30, 14] is an important methodology for differential privacy. Specifically, Laplace noise or Gaussian noise is added to data for protecting privacy. The privacy budget can be controlled by adjusting the scale of the noise. After normalization, the noisy labels are also biased soft labels in nature. In fact, given the probability density function of noise, we can calculate the corresponding unreliability degree and ambiguity degree in order to measure the effectiveness of the noisy labels. With the biased soft labels, the task is to train a utility model with strong privacy guarantees. Our theory can guarantee the utility of such soft labels.

To calculate unreliability degree and ambiguity degree, we refer to *order statistic* [8]. Order statistic analyze the $i$th-smallest value of random samples from a continuous distribution. We denote the *order distribution* $Order(d, n, i)$ as the $i$th-smallest value of $n$ samples from distribution $d$. The software Mathematics [45] provide an efficient API for estimating the order distribution.

**Corollary 2.** *Let $d$ denote the noise distribution (e.g. Laplace noise and Gaussian noise) and regard the noisy soft labels as the teacher. With the $k$ in Eq.1 and the total classes $c$, for $k \leq c - 1$, we can compute the $\eta$ and $\gamma$ as*

$$\eta = \Pr_{\substack{x \sim Order(d, c-1, n-k+1) \\ y \sim d}} (1 + y > x),$$

$$\gamma = \frac{\eta + k - 1}{c - 1}.$$

*With the same conditions in Theorem 2, the problem learning with additive noise is ERM learnable and the sample complexity remains unchanged.*

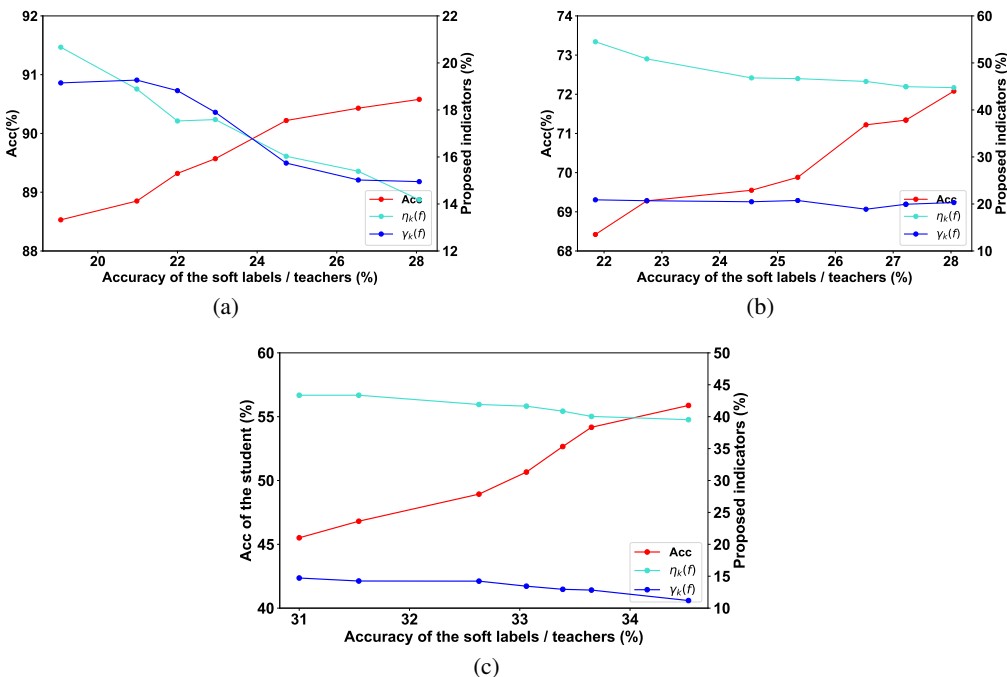

Figure 2: Indicators of soft labels generated by SBTs on CIFAR-10 (a), CIFAR-100 (b) and Tiny ImageNet (c). Acc represents the accuracy of the student model trained with the biased soft labels. Unreliability degree $\eta_k(f)$ and ambiguity degree $\gamma_k(f)$ are the two indicators proposed in our study.

As the scale of the noise increases, $\eta$ and $\gamma$ will increase, i.e., the effectiveness of the soft labels will decrease. This result agrees with the practical situation.

## 5 Experiments

Our experimentally investigate *whether biased soft labels are still effective* and *whether the proposed indicators can measure the effectiveness of these soft labels*, which consist of three parts. Firstly, we design a simple heuristic algorithm to generate Skillful but Bad Teachers (SBTs), which have low accuracy (less than 35%) but can teach good students. Secondly, experimental results demonstrate that students learning from SBTs can achieve comparable performance as models trained on the original data. We can explain these phenomenons with unreliability degree and ambiguity degree. Thirdly, we conduct experiments in weakly-supervised learning paradigms, and the results also confirm that unreliability degree and ambiguity degree can reflect the effectiveness of the biased soft labels. Due to space limitations, we present a part of experimental results in the appendix. Additionally, we provide the details of the experiment setup in appendix A.7.

### 5.1 Skillful but Bad Teachers

In this subsection, we introduced the design of the SBTs. The intuition behind SBTs, inspired by Theorem 2, is to inhibit correct predictions and reduces the unreliability degree and ambiguity degree simultaneously. So we design some heuristic loss functions and have some hyperparameters that qualitatively control the unreliability degree and ambiguity degree. More specifically, the designed loss functions are intended to keep the ground-truth label within the top-$k$ set of the soft labels, without necessarily requiring it to be at the top. Here, $k$ is an empirical constant, which we have set to 3 or 4 in training SBTs.

Firstly, SBTs will punish those correctly predicted instances as

$$\mathcal{L}_{\text{pun}}(\boldsymbol{x}, y) = -\mathbb{I}(\underset{j \in \mathcal{Y}}{\arg\max}(\boldsymbol{d}_j) = y)\ell(f(\boldsymbol{x}), y), \tag{5}$$

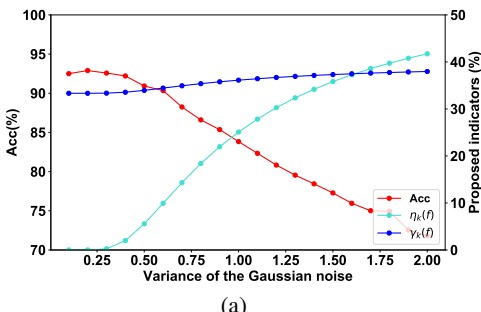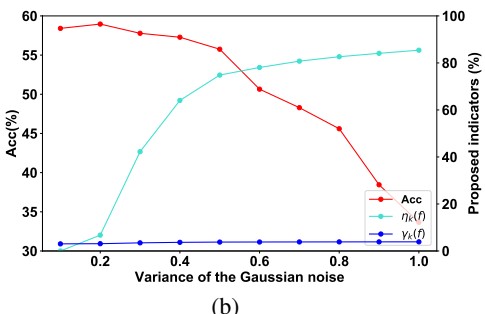

(a)                 (b)

Figure 3: Indicators of learning with Gaussian noise on CIFAR-10 (a) and CIFAR-100 (b).

where $\ell(\cdot, \cdot)$ is the cross entropy loss function. In practice, the value of the ground-truth label decreases significantly, which causes the ground-truth label to fall out of the top-$k$ set, i.e., large $\eta_k$. So the ground-truth label $y$ is compensated when $y \notin \Omega_k(f(\boldsymbol{x}))$:

$$\mathcal{L}_{\text{comp}}(\boldsymbol{x}, y) = \mathbb{I}(y \notin \Omega_k(f(\boldsymbol{x})))\ell(f(\boldsymbol{x}), y). \qquad (6)$$

The compensation term is designed to improve the top-$k$ accuracy of SBTs, which keeps the statistical effectiveness of the soft labels generated by SBTs. In practice, we found there was a strong correlation among the top-$k$ labels, which leaded to the confusion between the ground-truth label and similar labels, i.e., large $\gamma_k$. To decrease this correlation, we propose an effective method to make the labels in $\Omega_k(f(\boldsymbol{x}))$ as independent as possible. We randomly select $k - 1$ labels except the ground-truth label $y$. Then the selected $k - 1$ labels are employed as the learning objectives:

$$\mathcal{L}_{\text{rnd}}(\boldsymbol{x}, y) = \ell(f(\boldsymbol{x}), s_{\text{rnd}}). \qquad (7)$$

where $s_{\text{rnd}}$ is the set of the $k - 1$ random labels excluding $y$. Consequently, the total objective of SBTs is as follows:

$$\mathcal{L}(\boldsymbol{x}, y) = \mathcal{L}_{\text{ce}}(\boldsymbol{x}, y) + \alpha_1 \mathcal{L}_{\text{pun}}(\boldsymbol{x}, y) + \alpha_2 \mathcal{L}_{\text{comp}}(\boldsymbol{x}, y) + \alpha_3 \mathcal{L}_{\text{rnd}}(\boldsymbol{x}, y) \qquad (8)$$

where $\mathcal{L}_{\text{ce}}(\boldsymbol{x}, y)$ is the vanilla cross-entropy loss between the output of the model and the ground-truth label, and $\alpha_1, \alpha_2, \alpha_3$ are the trade-off parameters.

### 5.2 Effectiveness of the Proposed Indicators

In this subsection, our aim is to demonstrate that, despite the soft labels generated by SBTs are large-biased, students still achieve high accuracy. As outlined in Table 2 of the appendix, we adapt four different classic metrics: Chebyshev distance, KL divergence, Manhattan distance, and Euclidean distance. These metrics are employed to measure the discrepancies between the large-biased soft labels generated by SBTs and the ground-truth labels. The soft labels generated by SBTs exhibit substantial differences when compared to those from normally trained teachers. Nevertheless, they remain effective in instructing good students. It should be noted that the accuracy of the students mirrors the effectiveness of the soft labels. However, accuracy is an indirect measure contingent on the student model. The proposed unreliability degree and ambiguity degree can directly measure the effectiveness of these soft labels. The smaller the unreliability degree and ambiguity degree, the more effective the soft labels, and subsequently, the students' accuracy tends to be higher. This phenomenon is illustrated in Figure 2, which agrees with our expectations.

In the experiments, we set $k = 4$ for unreliability degree $\eta_k(f)$ and ambiguity degree $\gamma_k(f)$ ($k$ in the indicators can be different from $k$ in training SBTs). Since $\eta_k(f)$ and $\gamma_k(f)$ are probabilities over the entire data distribution $\mathcal{X} \times \mathcal{Y}$ and difficult to compute, we estimate them empirically by the the generated soft labels on the test data. Note that training with the ground-truth labels can achieve accuracy 95.29% on CIFAR-10, 78.13% on CIFAR-100 and 72.53% on Tiny ImageNet. We can find many interesting results in Figure 2:

- When these soft labels are generated by SBTs with accuracy less than 30%, which means they are quite different from the ground-truth labels, the students can still achieve accuracy much higher than the teachers.

- As $\eta_k(f)$ and $\gamma_k(f)$ decrease, Acc (i.e., accuracy of the student) increases. The proposed indicators exhibit an inverse correlation with the students' accuracy, which reflects the effectiveness of the proposed indicators.
- For the more complicated dataset like CIFAR-100 and Tiny ImageNet, accuracy (Acc) is more sensitive to $\eta_k(f)$ and $\gamma_k(f)$. This implies that, for more complex tasks, the quality of the annotation is more crucial.

The hyperparameters in Eq.(8) are pivotal in controlling the indicators of these soft labels, which is elucidated in A.8. Furthermore, we conducted experiments to investigate whether the effectiveness of soft labels is influenced by different model backbones, as detailed in A.9. We tested a range of architectures, including wideresnet 28x2, 28x4, 40x2, and 40x4. The four distinct backbones exhibited consistent patterns, indicating that the proposed indicators are effective across different backbones. In addition, we present the overall distribution of these soft labels in A.10. The experimental results demonstrate that *biased soft labels can also teach good students* and *the proposed indicators can measure the effectiveness of these soft labels*, which confirm the validity of our theory.

### 5.3 Biased Soft Labels in Weakly-Supervised Learning

The aforementioned weakly-supervised learning paradigms have engendered a lot of specialized algorithms. The intention of this paper is not to devise more efficacious algorithms, but to evaluate the effectiveness of the soft labels in the weakly-supervised learning paradigms by unreliability degree and ambiguity degree. The results of partial label learning are shown in Table 1, while the results of learning with additive noise are shown in Figure 3. Due to space limitations, the results of incomplete supervision are shown in A.11. From the three experiments, we can observe that the accuracy of the students decrease when unreliability degree and ambiguity degree increase. All the results reflect that these indicators can measure the effectiveness of the soft labels well, which is consistent with our theory. In addition, since the soft labels in partial label learning and learning with additive noise are generated by a mapping rule rather than a neural network, the results of these experiments are more stable and smooth.

Table 1: Students learning from partial labels.

| Dataset | $\gamma_k(f)$ | $\eta_k(f)$ | Student |
|---|---|---|---|
| CIFAR-10 | 0.1 | 0.1 | **93.98** |
| | 0.1 | 0.3 | **93.38** |
| | 0.1 | 0.5 | **91.94** |
| | 0.3 | 0.1 | **90.38** |
| | 0.3 | 0.3 | **88.59** |
| | 0.3 | 0.5 | **86.28** |
| | 0.5 | 0.1 | **85.11** |
| | 0.5 | 0.3 | **82.05** |
| | 0.5 | 0.5 | **77.95** |
| CIFAR-100 | 0.01 | 0.1 | **74.19** |
| | 0.01 | 0.3 | **73.16** |
| | 0.01 | 0.5 | **72.29** |
| | 0.05 | 0.1 | **68.08** |
| | 0.05 | 0.3 | **66.28** |
| | 0.05 | 0.5 | **63.69** |
| | 0.1 | 0.1 | **61.18** |
| | 0.1 | 0.3 | **58.6** |
| | 0.1 | 0.5 | **52.41** |

## 6 Conclusion

In this paper, we find that even large-biased soft labels can teach a good student and focus on the effectiveness of the biased soft labels. It motivates us to rethink when the biased soft labels (or teachers) are effective. We propose two indicators, unreliability degree and ambiguity degree, to measure the effectiveness of soft labels. Based on the proposed indicators, we provide moderate conditions that guarantee the classifier-consistency and ERM learnability of the biased soft label learning problem. Our theoretical framework can be adapted to elucidate the effectiveness of soft labels in three weakly-supervised learning paradigms, incomplete supervision, partial label learning and learning with additive noise. We design a heuristic method to train Skillful but Bad Teachers (SBTs), which validate that large-biased soft labels can teach good students. Besides, the effectiveness of both soft labels generated by SBTs and soft labels in the weakly-supervised learning paradigms can be measured by the proposed indicators well, which is consistent with our theory.

## Acknowledgement

This work was supported by the National Key Research & Development Plan of China (No. 2018AAA0100104), National Science Foundation of China (62125602, 62076063, 62206048), Natural Science Foundation of Jiangsu Province (BK20220819), and Young Elite Scientists Sponsorship Program of Jiangsu Association for Science and Technology Tj-2022-027.

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

# A   Appendix

## A.1   Variants of Knowledge Distillation

Self-distillation [55, 25, 50] treats the mixture of the outputs and the ground-truth labels as target soft labels, and the proportion is iteratively adjusted during the training. Ensemble KD [52, 60] employs the ensemble of the soft labels (teachers) to improve generalization. Besides, in mutual learning [56, 51], there is no explicit teacher network and multiple students learn from each other by synthesizing soft labels of other students. Soft labels play an important role in all these variants.

## A.2   Proof of Theorem 1

For simplification, we denote the probability of event $A$ as $p(A)$.

$$\mathrm{Err}_{\mathcal{D}}(h|f) = \mathbb{E}_{(\boldsymbol{x},y)\sim\mathcal{X}\times\mathcal{Y}}\Big[\min_{i\in\Omega_k(f(\boldsymbol{x}))}\ell(h(x),i)\cdot p(i\in\Omega_k(f(\boldsymbol{x}))) \mid f\Big]$$

$$= \mathbb{E}_{(\boldsymbol{x},y)\sim\mathcal{X}\times\mathcal{Y}}\Big[\min_{i\in\Omega_k(f(\boldsymbol{x}))}\{\ell(h(x),i)\cdot p(i\in\Omega_k(f(\boldsymbol{x})) \mid y\in\Omega_k(f(\boldsymbol{x})))\cdot p(y\in\Omega_k(f(\boldsymbol{x})) \mid f)$$
$$+ \ell(h(x),i)\cdot p(i\in\Omega_k(f(\boldsymbol{x})) \mid y\notin\Omega_k(f(\boldsymbol{x})))\cdot p(y\notin\Omega_k(f(\boldsymbol{x})) \mid f)\}\Big].$$

The coefficient of $\ell(h(x),y)$ is:

$$\mathrm{Coff}[\ell(h(x),y)] = p(y\in\Omega_k(f(\boldsymbol{x})) \mid y\in\Omega_k(f(\boldsymbol{x})))\cdot p(y\in\Omega_k(f(\boldsymbol{x})) \mid f)$$
$$= 1-\eta.$$

For $i\neq y$, there is

$$\mathrm{Coff}[\ell(h(x),i)] = p(i\in\Omega_k(f(\boldsymbol{x})) \mid y\in\Omega_k(f(\boldsymbol{x})))\cdot p(y\in\Omega_k(f(\boldsymbol{x})) \mid f)$$
$$+ p(i\in\Omega_k(f(\boldsymbol{x})) \mid y\notin\Omega_k(f(\boldsymbol{x})))\cdot p(y\notin\Omega_k(f(\boldsymbol{x})) \mid f)$$
$$\leq \gamma(1-\eta)+\eta.$$

Therefore, when $1-\eta > \gamma(1-\eta)+\eta$, i.e., $\gamma < 1 - \frac{\eta}{1-\eta}$, we have $h^* = \arg\min_{h\in\mathcal{H}} \mathrm{Err}_{\mathcal{D}}(h|f)$.

## A.3   Proof of Lemma 1

Let $\mathbf{z}, \mathbf{z}'$ be the training set and the testing set, and each of them is of size $n$. Define $H(\mathbf{z}) = \{h\in\mathcal{H} : \mathrm{Err}_{\mathbf{z}}^{\Delta}(h|f) = 0\}$ as the set of zero-empirical-risk hypotheses. We can bound $\Pr\left(\mathbf{z}\in R_{n,\varepsilon} \mid f\in H_{\eta}\right)$ with $\Pr\left((\mathbf{z},\mathbf{z}')\in S_{n,\varepsilon} \mid f\in H_{\eta}\right)$ as follows

$$\Pr\left((\mathbf{z},\mathbf{z}')\in S_{n,\varepsilon} \mid f\in H_{\eta}\right)$$
$$= \Pr\left((\mathbf{z},\mathbf{z}')\in S_{n,\varepsilon} \mid \mathbf{z}\in R_{n,\varepsilon}, f\in H_{\eta}\right)$$
$$= \Pr\left(\{\exists h\in H_{\varepsilon}\cap H(\mathbf{z}), \mathrm{Err}_{\mathbf{z}'}(h|f)\geq\frac{\varepsilon}{2}\} \mid \mathbf{z}\in R_{n,\varepsilon}, f\in H_{\eta}\right)$$
$$\geq \Pr\left(h\in H_{\varepsilon}\cap H(\mathbf{z}), \mathrm{Err}_{\mathbf{z}'}(h|f)\geq\frac{\varepsilon}{2} \mid \mathbf{z}\in R_{n,\varepsilon}, f\in H_{\eta}\right)$$
$$\geq 1-\exp\left(-\cdot\frac{\varepsilon n}{8}\right).$$

When $n > \frac{8\log 2}{\varepsilon}$, we have $\Pr\left((\mathbf{z},\mathbf{z}')\in S_{n,\varepsilon} \mid f\in H_{\eta}\right) \geq \frac{1}{2}\Pr\left(\mathbf{z}\in R_{n,\varepsilon} \mid f\in H_{\eta}\right)$, which completes the proof.

## A.4   Proof of Lemma 2

We follow the methodology of proving the ERM learnability in [26]. Here we need to bound $\Pr\left(S_{n,\varepsilon} \mid f\in H_{\eta}\right)$. The key behind the proof is to refine $\mathrm{Err}_{\mathbf{z}}^{\Delta}(h|f)$ and $\mathrm{Err}_{\mathbf{z}'}^{\Delta}(h|f)$. We use a classic method, i.e., swap, to refine the probability of a single instance. A swap $\sigma(\mathbf{z},\mathbf{z}') = (\mathbf{z}^{\sigma},\mathbf{z}'^{\sigma})$ means exchanging some instances between the training set $\mathbf{z}$ and testing set $\mathbf{z}'$ while keeping size $n$ unchanged. Each $(x,y)\in\mathbf{z}$ is a training instance while each $(x,y)\in\mathbf{z}'$ is a testing instance. There are $2^n$ different swaps in total and we define $G$ as the set of all swaps. Firstly, we use swap to describe $\Pr\left(S_{n,\varepsilon} \mid f\in H_{\eta}\right)$.

$$2^n \Pr\left((\mathbf{z}, \mathbf{z}') \in S_{n,\varepsilon} \mid f \in H_\eta\right) = \sum_{\sigma \in G} \mathbb{E}\left[\Pr((\mathbf{z}, \mathbf{z}') \in S_{n,\varepsilon} \mid x, y, x', y', f \in H_\eta)\right]$$

$$= \sum_{\sigma \in G} \mathbb{E}\left[\Pr(\sigma(\mathbf{z}, \mathbf{z}') \in S_{n,\varepsilon} \mid x, y, x', y', f \in H_\eta)\right]$$

$$= \mathbb{E}\left[\sum_{\sigma \in G} \Pr(\sigma(\mathbf{z}, \mathbf{z}') \in S_{n,\varepsilon} \mid x, y, x', y', f \in H_\eta)\right].$$

Here, the expectations are with respect to $(x, y, x', y')$. To further refine $S_{n,\varepsilon}$, we define $S_{n,\varepsilon}^h$ for a certain classifier $h$ as

$$S_{n,\varepsilon}^h = \{(\mathbf{z}, \mathbf{z}') : \mathrm{Err}_{\mathbf{z}}^\Delta(h|f) = 0, \mathrm{Err}_{\mathbf{z}'}(h|f) \geq \frac{\varepsilon}{2}\}.$$

Let $\mathcal{H} \mid (x, x')$ be the hypothesis space that have different prediction for instances $(x, x')$. Next, we have the bound

$$\sum_{\sigma \in G} \Pr(\sigma(\mathbf{z}, \mathbf{z}') \in S_{n,\varepsilon} \mid x, y, x', y', f \in H_\eta) \leq \sum_{h \in \mathcal{H} \mid (x, x')} \sum_{\sigma \in G} \Pr(\sigma(\mathbf{z}, \mathbf{z}') \in S_{n,\varepsilon}^h \mid x, y, x', y', f \in H_\eta).$$

By [33], the complexity of the hypothesis space $H \mid (x, x')$ can be bounded as

$$\left| \mathcal{H} \mid (x, x') \right| \leq (2n)^{d_H} L^{2d_H}.$$

Let $d_{x_i} = f(x_i)$ be the soft label of $x_i$ induced by model $f$ and $\Omega_k(d_{x_i})$ be the top-$k$ set of the soft label. Then,

$$\Pr\left(\sigma(\mathbf{z}, \mathbf{z}') \in S_{n,\varepsilon}^h \mid x, y, x', y', f \in H_\eta\right)$$

$$= \mathbb{I}\left(\mathrm{Err}_{\mathbf{z}'^\sigma}(h|f) \geq \frac{\varepsilon}{2} \mid f \in H_\eta\right) \cdot \Pr\left(\tilde{h}(x_i^\sigma) \in \Omega_k(d_{x_i}), 1 \leq i \leq n \mid x^\sigma, y^\sigma, f \in H_\eta\right)$$

$$= \mathbb{I}\left(\mathrm{Err}_{\mathbf{z}'^\sigma}(h|f) \geq \frac{\varepsilon}{2} \mid f \in H_\eta\right) \cdot \prod_{i=1}^n \Pr\left(\tilde{h}(x_i^\sigma) \in \Omega_k(d_{x_i}) \mid x^\sigma, y^\sigma, f \in H_\eta\right).$$

For the pair of $(x, y, x', y')$, we consider the number of instances of all cases. Specifically, let $u_1$, $u_2$ and $u_3$ represent the number of both incorrectly predicted instances, one incorrectly predicted instances and both correctly predicted instances. Besides, we define $u_\sigma$ as the number of instances where $(x^\sigma, y^\sigma)$ is incorrectly predicted while $(x'^\sigma, y'^\sigma)$ is correctly predicted. Afterwards, the number of incorrectly predicted instances in the testing set is $u_1 + u_2 - u_\sigma$.

$$\mathbb{I}\left(\mathrm{Err}_{\mathbf{z}'^\sigma}(h|f) \geq \frac{\varepsilon}{2} \mid f \in H_\eta\right) = \mathbb{I}(u_1 + u_2 - u_\sigma \geq \frac{\varepsilon}{2}n)$$

$$\leq \mathbb{I}(u_1 + u_2 \geq \frac{\varepsilon}{2}n).$$

For $\Pr\left(\tilde{h}(x_i^\sigma) \in \Omega_k(d_{x_i}) \mid x^\sigma, y^\sigma, f \in H_\eta\right)$, we count instances which have been swapped. On the one side, there are $u_2 + u_3 - u_\sigma$ instances are correctly predicted $(\tilde{h}(x_i^\sigma) = y_i^\sigma)$ where we have $\Pr\left(\tilde{h}(x_i^\sigma) \in \Omega_k(d_{x_i}) \mid f \in H_\eta\right) = 1 - \eta$ by (1). On the other side, there are $u_1 + u_\sigma$ instances are incorrectly predicted $(\tilde{h}(x_i^\sigma) \neq y_i^\sigma)$, where we have $\Pr\left(\tilde{h}(x_i^\sigma) \in \Omega_k(d_{x_i}) \mid f \in H_\eta\right) \leq \gamma$ by (2). So, we have

$$\prod_{i=1}^n \Pr\left(\tilde{h}(x_i^\sigma) \in \Omega_k(d_{x_i}) \mid x^\sigma, y^\sigma, f \in H_\eta\right) \leq (1 - \eta)^{u_2 + u_3 - u_\sigma} \cdot \gamma^{u_1 + u_\sigma}.$$

And then, the conditional probability can be bounded as follow:

$$\Pr\left(\sigma(\mathbf{z}, \mathbf{z}') \in S_{n,\varepsilon}^h \mid x, y, x', y'\right) \leq \mathbb{I}\left(u_1 + u_2 \geq \frac{\varepsilon}{2}n\right) \cdot (1 - \eta)^{u_2 + u_3 - u_\sigma} \cdot \gamma^{u_1 + u_\sigma}.$$

There are $2^n$ different swaps and we sum all.

$$\sum_{\sigma \in G} \mathbb{I}\left(u_1 + u_2 \geq \frac{\varepsilon}{2}n\right) \cdot (1-\eta)^{u_2+u_3-u_\sigma} \cdot \gamma^{u_1+u_\sigma}$$

$$\leq 2^{u_1+u_3} \cdot \mathbb{I}\left(u_1 + u_2 \geq \frac{\varepsilon}{2}n\right) \cdot \sum_{j=0}^{u_2} \binom{u_2}{j} (1-\eta)^{u_2+u_3-j} \cdot \gamma^{u_1+j}$$

$$= 2^{n-u_2} \cdot (1-\eta)^{u_2+u_3} \cdot \gamma^{u_1} \cdot \mathbb{I}\left(u_1 + u_2 \geq \frac{\varepsilon}{2}n\right) \cdot \sum_{j=0}^{u_2} \binom{u_2}{j} (\frac{\gamma}{1-\eta})^j$$

$$= 2^{n-u_2} \cdot (1-\eta)^{n-u_1} \cdot \gamma^{u_1} \cdot \mathbb{I}\left(u_1 + u_2 \geq \frac{\varepsilon}{2}n\right) \cdot (1+\frac{\gamma}{1-\eta})^{u_2}$$

$$= \mathbb{I}\left(u_1 + u_2 \geq \frac{\varepsilon}{2}n\right) \cdot 2^{n-u_2} \cdot (1-\eta)^{n-u_1} \cdot \gamma^{u_1} \cdot (\frac{1-\eta+\gamma}{1-\eta})^{u_2}$$

$$= \mathbb{I}\left(u_1 + u_2 \geq \frac{\varepsilon}{2}n\right) \cdot (2-2\eta)^n \cdot (\frac{\gamma}{1-\eta})^{u_1} \cdot (\frac{1-\eta+\gamma}{2(1-\eta)})^{u_2}.$$

where $n = u_1 + u_2 + u_3$. The $2^{u_1+u_3}$ in the first step means ways of swapping both correct instances and both incorrect instances. The index of $j$ equals $u_2$ different swaps of one correctly predicted instances. According to the assumption $\eta + \gamma \leq 1$, we have $0 < \frac{\gamma}{1-\eta} < \frac{1-\eta+\gamma}{2(1-\eta)} < 1$. For $u_1 + u_2 \geq \frac{\varepsilon}{2}n$, when $u_1 = 0$ and $u_2 = \frac{\varepsilon}{2}n$, the right side reaches its maximum as follows:

$$\Pr\left(S_{n,\varepsilon} \mid f \in H_\eta\right) \leq (2n)^{d_H} \cdot L^{2d_H} \cdot (2-2\eta)^n \cdot (\frac{1-\eta+\gamma}{2(1-\eta)})^{\frac{n\varepsilon}{2}}.$$

We have proved the Lemma 2.

### A.5  Proof of Theorem 2

With Lemma 1 and Lemma 2, we have

$$\Pr\left(R_{n,\varepsilon} \mid f \in H_\eta\right) \leq 2^{d_H+1} \cdot n^{d_H} \cdot L^{2d_H} \cdot (2-2\eta)^n \cdot (\frac{1-\eta+\gamma}{2(1-\eta)})^{\frac{n\varepsilon}{2}}.$$

We set $\theta$ as

$$\theta = \log\frac{2(1-\eta)}{1-\eta+\gamma}.$$

Since $\eta + \gamma < 1$, we get $\theta > 0$. We need to bound $\Pr\left(R_{n,\varepsilon} \mid f \in H_\eta\right)$ with $\eta$, which means

$$(d_H + 1) \cdot \log 2 + d_H \log n + 2d_H \log L + n\log\left(2-2\eta\right) - \frac{\theta\varepsilon n}{2} \leq \log\eta.$$

Note that the function $f(x) = \log\frac{1}{a} + ax - -1 \geq 0$. Let $a = \frac{\frac{\theta\varepsilon}{2} + \log(\frac{1}{2-2\eta})}{d_H}$ and $x = n$. It can be inferred that

$$\log n \leq \frac{\frac{\theta\varepsilon}{2} + \log(\frac{1}{2-2\eta})}{d_H}n - \log\frac{\frac{\theta\varepsilon}{2} + \log(\frac{1}{2-2\eta})}{d_H} - 1.$$

With the bound of $\log n$, we get the linear inequality of $n$. Since the aforementioned results hold for all $k \in \{1, 2, \ldots, c-1\}$, Let

$$n_0(\mathcal{H}, \varepsilon, \delta) = \min_{k \in \{1,2,\ldots,c-1\}}\left\{\frac{2}{\frac{\theta_k\varepsilon}{2} + \log\frac{1}{2-2\eta_k}}(d_{\mathcal{H}}(\log(2d_{\mathcal{H}})\right.$$

$$\left. + \log\frac{1}{\frac{\theta_k\varepsilon}{2} + \log\frac{1}{2-2\eta_k}} + 2\log L) + \log\frac{1}{\delta} + 1)\right\}.$$

When $n > n_0$, we get $\Pr\left(R_{n,\varepsilon} \mid f \in H_\eta\right) < \eta$ and the proof is finished.

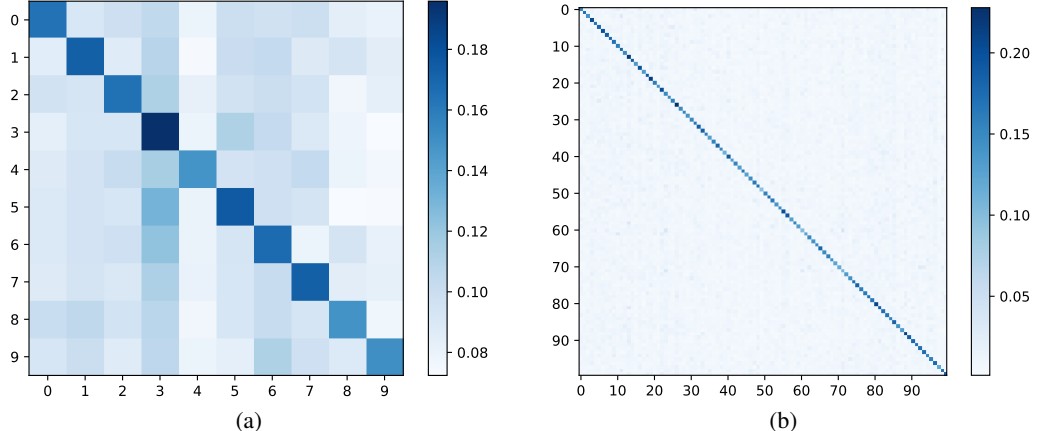

Figure 4: Average of the soft labels in CIFAR-10 (a) and CIFAR-100 (b).

### A.6 Proof of Theorem 3

Generally, for epoch $t$, we denote $\eta_t$ and $\gamma_t$ as the the unreliability degree and ambiguity degree of the the soft labels of the unlabeled data. The model will achieve the accuracy of epoch $t$, $\rho(\eta_t, \gamma_t)$. We refer to it as $\rho_t$ for simplicity. Then, for next epoch $t + 1$, the unreliability degree and ambiguity degree of next epoch can be estimated $\eta_{t+1} \leq 1 - \rho_t$. The estimation of ambiguity degree is a little bit more complicated. Incorrect label share equal probability $\gamma_{t+1}$ because we assume the noise is uniformly distributed. Then we have

$$\eta_{t+1} + (c - 1)\gamma_{t+1} = k.$$

Further,

$$\gamma_{t+1} \leq \frac{c - k - \rho_t}{c - 1}.$$

For $\rho(\eta, \gamma)$ monotonically decreases, with the upper bounds on $\eta_{t+1}$ and $\gamma_{t+1}$, we can get a lower bound on $\rho_{t+1}$ as

$$\rho_{t+1} = \rho(\eta_{t+1}, \gamma_{t+1})$$
$$\geq \rho(1 - \rho_t, \frac{c - k - \rho_t}{c - 1}).$$

If $\rho_{\text{final}} = \lim_{t \to \infty} \rho_t$ exists, it must satisfy the fix point equation,

$$x = \rho(1 - x, \frac{c - k - x}{c - 1}).$$

Next, we prove that if $\rho(\eta, \gamma)$ is $k_L$-*Lipschitz* continuous ($k_L < 1 - \frac{1}{c}$), then $\rho_{\text{final}}$ exists and is unique. We define

$$\psi : (\eta, \gamma) \to (1 - \rho(\eta, \gamma), \frac{c - k}{c - 1} - \frac{\rho(\eta, \gamma)}{c - 1}),$$

where $(\eta, \gamma) \in [0, 1]^2$. $l_1$-norm is employed as the norm on $[0, 1]^2$ and denote $d(\cdot, \cdot)$ as the distance function. We want to show that $\psi$ is a *contraction mapping*. For $(\eta_1, \gamma_1), (\eta_2, \gamma_2) \in [0, 1]^2$,

$$d(\psi(\eta_1, \gamma_1), \psi(\eta_2, \gamma_2))$$
$$= (1 + \frac{1}{c - 1})|\rho(\eta_1, \gamma_1) - \rho(\eta_2, \gamma_2)|$$
$$\leq (1 + \frac{1}{c - 1}) \cdot k_L d((\eta_1, \gamma_1), (\eta_2, \gamma_2))$$

where $(1 + \frac{1}{c-1}) \cdot k_L \in [0, 1)$. So $\psi$ is a *contraction mapping* and there is a unique fixed point $(\eta, \gamma)$ that $\psi(\eta, \gamma) = (\eta, \gamma)$. That means $\rho_{\text{final}}$ exists and is unique.

Table 2: Different metrics of soft labels.

| Dataset | Setting | Acc(Teacher) | Cheby | KL | Manhattan | Euclidean | Acc(Student) |
|---------|---------|--------------|-------|------|-----------|-----------|--------------|
| CIFAR-10 | Supervised | 95.29 | 0.0630 | 0.3335 | 0.1259 | 0.1728 | 95.68 |
| | SBTs | 19.07 | 0.8323 | 1.9295 | 1.6646 | 0.9093 | 88.53 |
| | | 20.98 | 0.8312 | 1.9619 | 1.6625 | 0.9129 | 88.85 |
| | | 22.00 | 0.8348 | 1.9197 | 1.6696 | 0.9078 | 89.32 |
| | | 22.95 | 0.8213 | 1.8781 | 1.6427 | 0.9008 | 89.57 |
| | | 24.73 | 0.8357 | 1.8968 | 1.6715 | 0.9058 | 90.22 |
| | | 26.54 | 0.8396 | 1.9110 | 1.6768 | 0.9076 | 90.43 |
| | | 28.07 | 0.8390 | 1.9099 | 1.6489 | 0.9150 | 90.58 |
| CIFAR-100 | Supervised | 78.13 | 0.1998 | 1.3763 | 0.3996 | 0.2630 | 79.01 |
| | SBTs | 21.85 | 0.9145 | 3.4739 | 1.8290 | 0.9467 | 68.42 |
| | | 22.74 | 0.9078 | 3.3814 | 1.8156 | 0.9391 | 69.28 |
| | | 24.55 | 0.9093 | 3.3457 | 1.8187 | 0.9392 | 69.55 |
| | | 25.35 | 0.9064 | 3.3299 | 1.8129 | 0.9370 | 69.88 |
| | | 26.53 | 0.9008 | 3.2921 | 1.8016 | 0.9320 | 71.22 |
| | | 27.22 | 0.9046 | 3.2858 | 1.8092 | 0.9329 | 71.34 |
| | | 28.05 | 0.9009 | 3.2560 | 1.8018 | 0.9302 | 72.08 |

## A.7 Experiment Setup

We consider three benchmark image datasets CIFAR-10, CIFAR-100 [23] and Tiny ImageNet, and generate the soft labels with different hyperparameters of the teacher model. The student model is trained with the generated soft labels and aims to distinguish the ground-truth label. The accuracy of the student model is employed to measure the effectiveness of the soft labels. Datasets are divided into training, validation, testing set in the ratio of 4:1:1. For the fairness of the experiments, all student models are WideResNet28×2 architecture [54] on each dataset. In all experiments, we use mini-batch SGD [36] with a batch size of 128 and a momentum of 0.9. Each model is trained with maximum epochs $T = 200$ and employs early stopping strategy with patience 20. We report final performance using the test accuracy corresponding to the best accuracy on validation set. For the all experiments, we employ a basic re-weighting strategy to train the student model. Specifically, in each epoch, we train the student model and update the soft labels with the softmax outputs of it.

**Weakly-supervised learning** For PLL, we employ a common strategy to zeroize the soft labels that are not in the candidate set. For learning with unlabeled data, we warm-up the model for 5 epochs. Then, we train with all data every epoch and label the unlabeled data with the model every 5 epochs. Each dataset is divided into labeled set, unlabeled set, testing set in the ratio of 1:19:4.

## A.8 Hyperparameters in (8)

We set the number of the random labels (in 7) as 3 while $k$ in $\eta_k$ and $\gamma_k$ as 4 in all experiments. The punishment factor $\alpha_1$ ranges from 0 to 0.4 and the compensation factor $\alpha_2$ ranges from 0.9 to 1.3. The weight of the random labels $\alpha_3$ is from 1.6 to 2.3. In detail, we observe the following phenomenons:

- As mentioned in 1, we cannot take the accuracy of the teacher as the sole criterion to evaluate the pedagogical efficacy. $\gamma$ defined in (2) can be seen as the coarse measure of the imbalance of the label. For the student model, the smaller $\gamma$ could mean the better performance if the number of random labels is fixed.

- Both the punishment and the random labels are applied to decrease the top-1 accuracy within a reasonable range. For the simple dataset, large punishment are needed to protect privacy and for the complicated dataset like CIFAR-100, we enhance the ratio of random labels to reduce the effect of similar labels.

- These soft labels are very different from the ground-truth labels, which means low accuracy. In these soft labels, the average degree of the ground-truth labels is around 0.2.

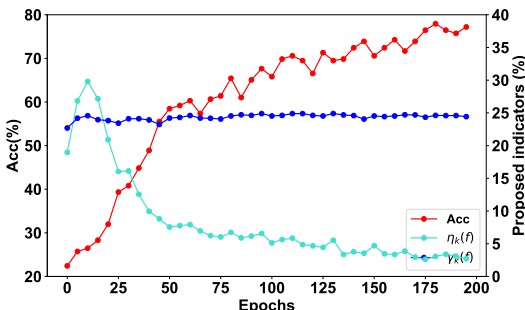

Figure 5: The curve of the indicators in incomplete supervision.

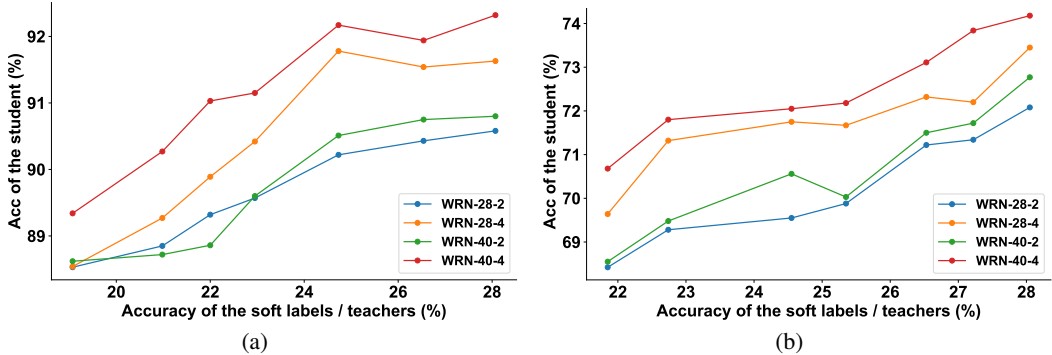

| (a) | (b) |
|---|---|

Figure 6: The experiments of 4 different backbones: WRN 28×2, WRN 28×4, WRN 40×2, WRN 40×4. On the left are the experiments conducted on CIFAR-10, while on the right are those performed on CIFAR-100.

## A.9 Experiments of different backbones

Figure 6 demonstrates whether the effectiveness of soft labels is influenced by different backbones. We experimented with wideresnet 28x2, 28x4, 40x2, and 40x4. In the Figure 2, we omitted the Unreliability degree and Ambiguity degree, which are the same the that of Figure 2 in the paper, for clarity. The four distinct backbones displayed consistent trends, further suggesting that the proposed indicators are effective across different backbones.

## A.10 Overall distribution of the Soft Labels Generated by SBTs

As illustrated in Figure 4, the horizontal axis represents the ground-truth label while the vertical axis represents the mean of the soft labels. The diagonal can be seen as the degree of correctly predicted labels. We can see that the ground-truth label is dominant in the soft labels. On the other side, the figure can be seen as a simple measure of the similarity between labels.

## A.11 Experiments of Incomplete Supervision

The predictive model label the unlabeled data and then learn with all data alternately. As shown in Figure 5, Acc (the accuracy of the model) improves as the soft labels evolve. Accordingly, unreliability degree $\eta_k(f)$ decreases and ambiguity degree $\gamma_k(f)$ remain unchanged, which means the soft labels of the model are more effective. The figure shows the dynamics in the training with unlabeled data, which is consistent with the theory in 4.1. Furthermore, we find that the soft labels of unlabeled data exhibit a high ambiguity degree $\gamma_k(f)$, indicating that some categories may be very similar and difficult for the model to distinguish.

