# OpenReview forum: "Learning From Biased Soft Labels"
_NeurIPS.cc/2023/Conference — NeurIPS 2023 poster_

### Official Review · Reviewer_HDpR · 2023-06-30

**Soundness:** 4 excellent
**Presentation:** 3 good
**Contribution:** 3 good
**Rating:** 6
**Confidence:** 3

**Summary:**

The authors analyse the learnability properties of (biased) soft labels, e.g., originating from a teacher model in a student-teacher setup, with respect to classifier consistency and ERM learnability. To this end, two indicator measures of the quality of the proxy labels are suggested, which are used in a theoretical analysis to derive bounds about the aforementioned quality dimensions. Moreover, a heuristic loss for training skillful but bad teachers is developed.

**Strengths:**

- Investigates a highly relevant matter
- Technically sound and valid theoretical results
- Covers multiple special cases of weakly-supervised learning as a practically relevant problem setup


**Weaknesses:**

Major:
- I find the distinction between incomplete supervision and partial label learning confusing. Typically, incomplete supervision and partial label learning are more or less the same thing. What the authors here refer to in the case of incomplete supervision is semi-supervised learning, which is a special case of partial label learning, namely that you can either observe a precise (i.e., unambiguous) label, or no label at all resp. the complete target space. In order to preserve consistency with related literature on this (e.g., as in your reference [58]), I would recommend sticking to this ontology.
- The empirical analysis of SBTs does not contain any baseline. Overall, it is hard to judge the appropriateness of the developed indicators and the SBT loss proposals, perhaps the experimental setup can be overhauled.
- It is hard to assess the looseness resp. tightness of the bound in Theorem 2. I would love to see the authors elaborating on this matter, e.g., by putting individual term components in a context, comparing it to “classical” bounds in multi-class learning, potentially with respect to label noise.

Minor:
- The page limit for submissions was 9 pages, but there is content on page 10.
- I think there is a missing $\max$ (or a different aggregator) in Eq. (1), right?
- Also, “ambiguity degree” is a term that is already being used in the PLL literature, as also indicated by referring to [5]. This is not precisely an equivalent ambiguity degree formulation, so it should be distinguished.


**Questions:**

- How did you determine the hyperparameters being used in the experiments? Did you repeat the experiments multiple times with different seeds? How about the standard deviations?

**Limitations:**

Yes.

---

> ### Author Rebuttal · Authors · 2023-08-09
>
> Dear Reviewer,
>
> We are truly grateful for your comprehensive review and the positive recognition of our work's relevance, technical soundness, and valid theoretical results. Your feedback provides us with valuable insights that will undoubtedly enhance the quality of our paper. We would like to address the concerns you raised:
>
> **Q1. Incomplete supervision and partial label learning.**
>
> * A: Incomplete supervision (semi-supervised learning) is training with labeled data and unlabeled data (i.e. the dataset are partially labelled). In partial label learning, each instance is ambiguously labeled with a labels set and the ground-truth label is in the set (i.e. each instance are partially labelled). There is not a direct correlation between incomplete supervision and partial label learning.
>
> **Q2. The baseline of the empirical analysis of SBTs.**
>
> * A: Our primary contribution is the discovery that large-biased soft labels can also produce competent student models and that we provide explanations and theoretical proofs for the underlying phenomenon. The algorithm we designed (see Section 5.1) primarily serves to validate our theory. To the best of our knowledge, this phenomenon hasn't been studied previously, which is why we did not design comparative experiments.
>
> **Q3. The looseness resp. tightness of the bound in Theorem 2.**
>
> * A: Suppose the Natarajan dimension of the hypothesis space $\mathcal{H}$ is $d_\mathcal{H}$.
> In the multi-class learning, a classic sample complexity for the ERM learners is $\mathcal{O}(\frac{d_{\mathcal{H}} \log \frac{1}{\varepsilon} + \log \frac{1}{\delta} }{\varepsilon})$ [1]. This bound is based on a fully supervised scenario, whereas weakly supervised problems introduce a great deal more complexity. In this context, we refer to a theory on partial label learning [2] and describe its bound based on the proposed indicators: $\mathcal{O}(\frac{d_{\mathcal{H}} \log d_{\mathcal{H}} + \log \frac{1}{\theta \varepsilon} + \log \frac{1}{\delta} }{\theta \varepsilon})$, where $\theta = \log \frac{2}{1+\gamma}$. The bound in Theorem 2 (ours) is $\mathcal{O}(\frac{d_{\mathcal{H}} \log d_{\mathcal{H}} + \log \frac{1}{\theta \varepsilon} + \log \frac{1}{\delta} }{\theta \varepsilon})$, where $\theta = \log \frac{2(1-\eta)}{1-\eta+\gamma}$. The bound in Theorem 2 is of the same order of magnitude as that in [2]. The difference between the two lies in the value of $\theta$. When the unreliability degree $\eta>0$, more samples are required, almost increasing in proportion to $\frac{1}{\theta}$.
>
>   >[1] Ben-David, Shai, Nicolo Cesa-Bianchi, and Philip M. Long. "Characterizations of learnability for classes of {O,…, n}-valued functions." Proceedings of the fifth annual workshop on Computational learning theory. 1992.
>   >[2] Liu, Liping, and Thomas Dietterich. "Learnability of the superset label learning problem." International Conference on Machine Learning. PMLR, 2014.
>
> **Q4. Exceed the page limit.**
>
> * A: We sincerely apologize for this oversight. We will make the necessary adjustments to condense the length of our paper.
>
> **Q5. Is there a missing $\max$ (or a different aggregator) in Eq. (1)?**
>
> * A: Thank you for your kind reminder. We have revisited and reflected upon the definition of $\eta$.
> Aggregation based on inter-class max is better, with $\eta= max_{(\boldsymbol{x}, y) \sim \mathcal{X} \times \mathcal{Y}} \operatorname{Pr}(y \notin \Omega_k(f(x)))$. The form in Eq. (1) implies that all samples have the same unreliability degree, which is right but more stringent.
>
> **Q6. The term “ambiguity degree” in the PLL literature should be distinguished.**
>
> * A: As explained in lines135-136, the ambiguity degree is inspired by PLL and extends it to soft labels. In fact, if PLL is transformed into the form of soft labels, two kinds of ambiguity degree are identical. To be more precise, the ambiguity degree in this paper refers to the ambiguity degree of soft labels.
>
> **Q7. How to choose hyperparameters?**
>
> * A: We have provided a detailed description of our experimental setup in Appendix A.7. Additionally, we've outlined the range and rationale behind our hyperparameter design in Appendix A.9.
>
> **Q8. Repeat the experiments multiple times with different seeds.**
>
> * A: Due to the limited time for the rebuttal, we replicated our experiments on CIFAR10/CIFAR100 using three different seeds. In each experiment, while using the same teacher, we varied the seed during student training. As a result, we observed a standard deviation of 1.113 in accuracy on CIFAR10 and 1.215 on CIFAR100.
>
> Your constructive feedback is instrumental in refining our paper, and we are committed to making the necessary improvements. Once again, thank you for your valuable insights and for considering our paper for acceptance.

---

> > ### Comment · Reviewer_HDpR · 2023-08-13
> >
> > Thanks for your efforts and the thorough response. I am now more certain about going for an accept, which is why I increased my score.
> >
> > One last remark on the terms "incomplete supervision" vs. "partial labels": Pointing again to the reference [58], I find it more consistent with classical weakly-supervised learning (WSL) ontologies when "incomplete supervision" is an abstract term for learning settings, where not all labels are unambiguously labeled. E.g., in classification problems with a target space $\mathcal{Y}$, one wouldn't necessarily observe only labels $y \in \mathcal{Y}$, but *at least* one instance with a label $Y \subseteq \mathcal{Y}$ with $|Y|>1$. Semi-supervised learning with a labeled and unlabeled split would then refer to instances labeled with a single $y \in \mathcal{Y}$ (labeled split) and $Y=\mathcal{Y}$ (unlabeled), i.e., it considers the extreme case of observing only deterministic and agnostic "partial" labels. Partial label learning in general is more abstract in not specifying how the partial labels $\subseteq \mathcal{Y}$ are observed in the data, but typically refer to "mixed" cases where we observe something in between the two extremes of $\mathcal{Y}$ and $y$. Incomplete supervision would merely refer to this more abstract term, at least from the point of how it is used within the WSL community. But that is more of a minor remark.

---

> > > ### Author Response · Authors · 2023-08-13
> > >
> > > We are truly appreciative that our response has garnered your approval, and we would like to express our gratitude once again for the elevated rating you have provided. Your support serves as a significant source of encouragement for our work.
> > >
> > > The lack of clarity about "incomplete supervision" and "partial labels" in our paper led to your misunderstanding. We take full responsibility for this oversight, and it is not indicative of any shortcomings on your part. The term "partial" indeed has the potential for ambiguity. To rectify this, we will include further elucidation of both concepts within the main body of the text.
> > >
> > > Thank you for your insightful feedback and understanding.

---

### Official Review · Reviewer_vNqZ · 2023-07-05

**Soundness:** 3 good
**Presentation:** 3 good
**Contribution:** 2 fair
**Rating:** 5
**Confidence:** 3

**Summary:**

This paper studies the effectiveness of biased soft labels in knowledge distillation and weakly-supervised learning. The paper introduces two indicators to measure the effectiveness of soft labels, and proposes moderate conditions to ensure that biased soft label learning is classifier-consistent and ERM learnable. The paper also presents a heuristic method to train skillful but bad teachers, and shows that they can teach students to achieve high accuracy on CIFAR-10/100. The paper applies the theoretical framework to three weakly-supervised learning paradigms, and validates the indicators with experiments.

**Strengths:**

1. The paper is well-written and a pleasure to read.
2. The paper provides thorough theoretical analysis for three different weak supervision settings.

**Weaknesses:**

1. The paper only conducts experiments on CIFAR-10/100 datasets, and does not provide experiments on larger and mainstream datasets, such as ImageNet. The paper also does not compare the proposed method with other state-of-the-art weakly-supervised methods, especially those based on knowledge distillation.
2. The paper is suggested to provide more visualizations of the the prediction results (class probabilities) of large-biased soft labels (generated by SBTs) and good student.
3. The paper does not provide experiments on different backbones, and cannot demonstrate the effectiveness of the proposed indicators and method on different architectures. The paper also does not discuss how the choice of the backbone affects the performance and robustness of the method.
4. The paper exceeds the page limit. The paper is suggested to be shortened to meet the page limit requirement.

**Questions:**

1. In the introduction, the paper claims that “Empirical Risk Minimization (ERM) learners’ performance can generalize to the entire data distribution.”
    However, previous studies have pointed out the drawbacks of ERM, such as (1) neural networks trained with ERM change their predictions drastically when evaluated on examples just outside the training distribution[1], also known as adversarial examples; (2) ERM allows large neural networks to memorize (instead of generalize from) the training data even in the presence of strong regularization, or in classification problems where the labels are assigned at random [1].
    In general, using ERM for optimization may affect the generalization ability of the model. Does this contradict the conclusion of this paper? Does the conclusion of this paper have any prior studies or experiments to support it?
[1]Zhang, Hongyi, et al. "mixup: Beyond empirical risk minimization." *arXiv preprint arXiv:1710.09412* (2017).

2. The paper mentioned “skillful but bad teachers”. How to define the skillfulness of the bad teachers?
3. As you mentioned in Section 5.3, is your main contribution the two methods of evaluating soft labels (unreliability degree and ambiguity degree), rather than the heuristic method of “bad teacher can teach good student”?
4. The conclusion that “the accuracy of the students decrease when unreliability degree and ambiguity degree increase” does not seem to be obvious in Figure 2(b). Also, in Figure 3 (a) and (b), the ambiguity degree γ does not change much as the student accuracy increases. The same situation also occurs in Figure 5 in the appendix, which seems to not support the effectiveness of the ambiguity degree γ indicator in these weakly-supervised settings.

**Limitations:**

Some of the paper’s experimental results show that the two indicators proposed by the paper cannot well characterize the effectiveness of soft labels in all weakly-supervised settings. For example, under some of the weakly-supervision settings, the paper shows that the indicators are not consistent with the performance of the student model. The paper does not provide sufficient explanation or analysis for this phenomenon, and does not discuss how to improve or modify the indicators to better capture the effectiveness of soft labels.

---

> ### Author Rebuttal · Authors · 2023-08-09
>
> Dear Reviewer,
>
> **Q1. Experiments on Tiny-ImageNet.**
>
> - A: We have added experiments on Tiny-Imagenet, which can be found in the PDF attached to the rebuttal.
>
> **Q2. Why not compare the proposed method with other state-of-the-art weakly-supervised methods.**
>
> - A: Our primary contribution is the discovery that large-biased soft labels can also produce competent student models and that we provide explanations and theoretical proofs for the underlying phenomenon. The algorithm we designed (see Section 5.1) primarily serves to validate our theory. The aim of the weakly-supervised experimentation was not to train the state-of-the-art student model. To the best of our knowledge, this phenomenon hasn't been studied previously, which is why we did not design comparative experiments.
>
> **Q3. Visualizations of the class probabilities of large-biased soft labels.**
>
> - A: We have provided the visualizations of the class probabilities of large-biased soft labels (generated by SBTs) in the appendix (see Figure 4). We did not include it in the main text due to space constraints.
>
> **Q4. Experiments on different backbones.**
>
> - A: We have added experiments on different backbones (Figure 2 in the PDF of Author Rebuttal).  We experimented with wideresnet 28x2, 28x4, 40x2, and 40x4. In the Figure 2, we omitted the Unreliability degree and Ambiguity degree (which is the same the that of Figure 2 in the paper) for clarity. The four distinct backbones displayed consistent trends, further suggesting that the proposed indicators are effective across different backbones. For the same teacher model, the better the student model's fitting capacity, the better student's performance tends to be.
>
> **Q5. The paper exceeds the page limit.**
>
> - A: We sincerely apologize for this oversight. We will make the necessary adjustments to condense the length of our paper.
>
> **Q6. ERM learner cannot generalize well.**
>
> - A: The expected classification error defined in section 3.1 indicates the performance (or generalization ability) of the model with respect to the original data distribution. This conclusion is widely accepted in the machine learning community, as evidenced by [1,2]. Peter L. Bartlett also stated, 'The performance of such a model selection scheme critically depends on how well the error bounds match the true error (i.e., expected classification error).' As for the model's performance on adversarial examples, it falls outside the scope of our study. The generalization ability mentioned in mixup refers to situations outside the training distribution, and there's no contradiction between the two. Thank you for the excellent idea regarding the student model's performance on adversarial examples, and we will consider the feasibility of this direction in our future work.
>
>   > [1] Bartlett, Peter L., and Shahar Mendelson. "Rademacher and Gaussian complexities: Risk bounds and structural results." Journal of Machine Learning Research 3.Nov (2002): 463-482.
>   > [2] Daniely, Amit, et al. "Multiclass learnability and the erm principle." Proceedings of the 24th Annual Conference on Learning Theory. JMLR Workshop and Conference Proceedings, 2011.
>
> **Q7. Definition of skillfulness of the bad teachers.**
>
> - A: We apologize for any confusion caused by the term "skillfulness of the bad teachers" mentioned in the paper. Here, we provide a rigorous definition for "skillfulness of the bad teachers" to clarify its meaning.
>
>   > Definition 1. (Bias of soft labels) Give a dataset $D$ consisting of $n$ samples, the feature vector for the $i$-th sample is denoted as $\boldsymbol{x}_i$ and the corresponding label is denoted as $y_i$. Let $f$ represent a model or a mapping rule (e.g. label smoothing). The bias of the soft labels generated by $f$ on dataset $D$ is
>
>   \begin{equation}
>   Bias(f, D)=\frac{1}{n} \sum_{i=1}^n [1-f_{y_i} (\boldsymbol{x}_i)],
>   \end{equation}
>
>   where $f_{y_i} (\boldsymbol{x}_i)$ refers to the component of the soft label $f(\boldsymbol{x}_i)$ that corresponds to the true label $y_i$.
>
>   > Definition 2. (Large-biased soft labels) Soft labels generated by $f$ on dataset $D$ is called biased soft labels when $Bias(f, D) \textgreater 0$ and called large-biased soft labels when $Bias(f, D) \geq 0.5$.
>
>   > Definition 3. (Bad teachers) We define $f$ as a bad teacher if the soft labels it generates on dataset $D$ are large-biased. Typically, $D$ is the training set for $f$.
>
> - We cannot provide a precise definition for 'skillful teachers' as the performance of the student is contingent upon the architecture of the model and the complexity of the dataset. In this context, 'skillful' merely signifies that the teacher produces students with acceptable outcomes.
>
> **Q8. Main contribution.**
>
> - A: We discover that 'large-biased soft labels can produce competent student models' and then introduced two indicators (unreliability degree and ambiguity degree) and provided theoretical guarantees for them. The heuristic approach of 'bad teacher can teach good student' serves to empirically validate the existence of the phenomenon and the soundness of our theory.
>
> **Q9. The conclusion that “the accuracy of the students decrease when unreliability degree and ambiguity degree increase” does not seem to be obvious.**
>
> - A: We sincerely apologize for any confusion caused by our statement 'the accuracy of the students decrease when unreliability degree and ambiguity degree increase'. What we intended to convey was that when one remains unchanged and the other increases, the accuracy of the students will decrease. Both indicators together are necessary to assess the efficacy of soft labels, and one should not be analyzed based solely on individual trends. This applies to Figure 2(b), 3(a), and 3(b) alike.
>
> Thank you for your valuable feedback. We will make revisions to our paper based on your suggestions.

---

> > ### Author Response · Authors · 2023-08-18
> >
> > Dear Reviewer
> >
> > Could we kindly know if the responses have addressed your concerns and if further explanations or clarifications are needed? Your time and efforts in evaluating our work are appreciated greatly.

---

> > ### Comment · Reviewer_vNqZ · 2023-08-18
> >
> > Thank you for your reply and rebuttal to my comments.
> >
> > I appreciate the authors' detailed responses, which have addressed most of my concerns effectively. In particular, I appreciate the addition of new experiments and the clarification of previously vague concepts in the paper.
> >
> > Based on these improvements and responses, I am pleased to give the paper a positive evaluation and will increase my score to 5. The authors are supposed to incorporate the insights from our discussion into the revised version of the paper.

---

### Official Review · Reviewer_mHw2 · 2023-07-07

**Soundness:** 2 fair
**Presentation:** 2 fair
**Contribution:** 3 good
**Rating:** 5
**Confidence:** 3

**Summary:**

The authors propose a clever theoretical framework for studying learnability under supervision with imperfect soft labels.

**Strengths:**

- Learning from imperfect soft labels is an important and interesting area that is prevalent not only in knowledge distillation but also in cognitive science, human-AI interaction, and pretty much any modeling that aim to take into account noise and uncertainty.
- The authors conduct a promising theoretical analysis of this setting by recasting it as a noisy top-k oracle problem (where teachers provide a set of k labels which with some probability contains the true label).

**Weaknesses:**

- Clarity could be improved by providing more intuitions and explanations for terminology throughout. For example, "biased soft label" is never formally defined, and the implied definition (based on the notions of ambiguity and unreliability) does not fully agree with what I would normally think of when thinking of statistical notions of bias. Some additional editorial revision (fixing typos, etc.) would also be helpful, but I am not taking that into account in my score.
- I think the assumptions are not quite as moderate as the authors imply and would like some more clarification about these. For example, one of the assumptions seem to be that labels occur in the top-k with equal probability, but this is clearly not the case in practice (e.g. you may believe a picture of a dog could be a wolf, but certainly not an airplane) which the authors mention when designing their experiments as they find that the top-k labels are correlated. The experiments are then designed to satisfy this assumption which means there is very little evidence that the proposed theory describes a realistic setting.  Another assumption (Assumption 2 in 4.1) seems to pre-assume that the metrics considered by the authors are inversely correlated with accuracy which feels circular since I believe this is one of the things the authors want to claim.
- As mentioned in the section above, the empirical results are currently unconvincing. In addition to the assumption mentioned above, it seems like in both experiments, the true label has the highest probability in >15% of the cases. For CIFAR10 this is reasonable since random chance is 10%, but for CIFAR100, this is quite high. Is there a disconnect between the theory setting and the experimental setting, in that the theory is focused on learning from the top-k set where all top-k labels are treated equally while in the experimental setting, learning still takes into account the actual probabilities? Perhaps a more compelling experiment would be to isolate various top-k cases and show that the curves are robust to changes in k?
- There are a number of existing papers that study the informativeness of soft labels. It would be great to see some discussion of how this study fits into that literature.

**Questions:**

See weaknesses section.

**Limitations:**

- The authors have listed both limitations and assumptions, though I think the assumptions may be more of a limitation than suggested.

---

> ### Author Rebuttal · Authors · 2023-08-08
>
> Dear Reviewer,
>
> **Q1. Clairty and defination.**
>
> - A: We apologize for any confusion caused by the term "large-biased soft labels" mentioned in the paper. Here, we provide a rigorous definition for "large-biased soft labels" to clarify its meaning.
>
>   > Definition 1. (Bias of soft labels) Give a dataset $D$ consisting of $n$ samples, the feature vector for the $i$-th sample is denoted as $\boldsymbol{x}_i$ and the corresponding label is denoted as $y_i$. Let $f$ represent a model or a mapping rule (e.g. label smoothing). The bias of the soft labels generated by $f$ on dataset $D$ is
>
>   \begin{equation}
>   Bias(f, D)=\frac{1}{n} \sum_{i=1}^n [1-f_{y_i} (\boldsymbol{x}_i)],
>   \end{equation}
>
>   where $f_{y_i} (\boldsymbol{x}_i)$ refers to the component of the soft label $f(\boldsymbol{x}_i)$ that corresponds to the true label $y_i$.
>
>   > Definition 2. (Large-biased soft labels) Soft labels generated by $f$ on dataset $D$ is called biased soft labels when $Bias(f, D) \textgreater 0$ and called large-biased soft labels when $Bias(f, D) \geq 0.5$.
>
> **Q2. About the assumption "labels occur in the top-k with equal probability".**
>
> - A: The assumption "incorrect labels occur in the top-k with equal probability", mentioned in line 212 (Section 4.1), is solely in service of Theorem 3. I have also articulated that this assumption can be relaxed, mentioned in lines 213-215, for instance, by considering an upper bound of $\frac{p(i|x)}{p(j|x)}$. This does not impact our understanding of Incomplete Supervision from the perspective of soft labels.
>
> **Q3. The designed experiments reflect the assumption is not realistic.**
>
> - A: In fact, for models that are trained normally, the conditions we described (i.e. $\gamma_k(f)<1-\frac{\eta_k(f)}{1-\eta_k(f)}$ and $\eta_k + \gamma_k<1$) are easily met. It only requires the existence of a specific 'k' such that both delta and gamma are satisfied. To further illustrate this point, when we set $k=4$ and use a normally trained resnet-50 as an example, we obtained the following results:
>
>   - CIFAR10: Accuracy (ACC): 95.29%; $\eta$: 2.63\%; $\gamma$: 20.12\%.
>   - CIFAR100: Accuracy (ACC): 78.13%; $\eta$: 6.55; $\gamma$: 21.01\%.
>   - Tiny-Imagenet: Accuracy (ACC): 60.63%; $\eta$: 13.55; $\gamma$: 24.67\%.
>
>   They all easily satisfy the requirements stipulated in Theorem 1 and Theorem 2.
>
> **Q4. (Assumption 2 in 4.1) Assume that the metrics are inversely correlated with accuracy.**
>
> - A: In Assumption 2, we indeed ideally posit a inverse correlation between the proposed metrics and accuracy, using this to prove Theorem 3. Operating under the ideal assumption, we can justifiably explain both the dynamic process and the final performance of Incomplete Supervision. This indicates that our theory does not contradict practical observations, further validating the reasonableness and practicability of our theoretical framework.
>
> **Q5. The accuracy of the teacher is much higher than 1% on CIFAR100.**
>
> - A: Theoretically, the accuracy on CIFAR100 can be even lower. However, in practice, the accuracy being above 15% is constrained by the size of the dataset. As seen in Theorem 2, the smaller the unreliability degree, the larger the required dataset size. If there are more CIFAR100 training samples, we believe the accuracy could indeed be lower.
>
> **Q6. Top-k labels and the actual probabilities of the soft labels.**
>
> - A: It's important to clarify that our study consistently focuses on learning from soft labels, rather than from top-k sets. The indicators we proposed facilitate theoretical analysis by transforming soft labels into top-k sets. However, the conclusions are universally valid for all soft labels. While in practice experiments are indeed influenced by the “actual probabilities,” our theory merely provides a guarantee in the worst-case scenario.
>
> **Q7. About the choice of k.**
>
> - A: It's evident that when k=1 or k=c−1 (where c is the total number of classes), the supervision information suffers significant loss, resulting in poor performance by the student model. In practice, we observed that the student performs well when k=3, 4, 5. Our experimental results are based on a fixed k=4.
>
> **Q8. How this study fits into related literature.**
>
> - A: To the best of our knowledge, teacher models in the current domains of knowledge distillation and label enhancement aim to mimic true labels while adding some regularization terms. Such teacher models typically produce soft labels that closely resemble the true labels. There are many existing explanations for these types of soft labels, such as they act as a form of regularization, they approximate Bayesian prior probabilities, they prevent excessive overconfidence, etc (see lines 80-84, 100-101). However, it seems that these theories cannot account for the observation that "large-biased soft labels can also work". Our research serves as a complement and refinement to the existing theories on soft labels.
>
> Thank you for your valuable feedback. We will make revisions to our paper based on your suggestions.

---

> > ### Author Response · Authors · 2023-08-18
> >
> > Dear Reviewer
> >
> > Could we kindly know if the responses have addressed your concerns and if further explanations or clarifications are needed? Your time and efforts in evaluating our work are appreciated greatly.

---

> > ### Comment · Reviewer_mHw2 · 2023-08-20
> >
> > Thank you for the detailed rebuttal!
> > My concerns have generally been addressed and I have updated my score accordingly.

---

### Official Review · Reviewer_M44N · 2023-07-07

**Soundness:** 3 good
**Presentation:** 3 good
**Contribution:** 3 good
**Rating:** 6
**Confidence:** 4

**Summary:**

This paper aims to study the effectiveness of biased soft labels for knowledge distillation. They propose two indicators to measure the effectiveness of the biased soft labels: unreliability degree and ambiguity degree.  They provide a theoretical guarantee that the biased soft labels are effective in training a good student in three weakly-supervised learning paradigms: incomplete supervision, partial label learning, and learning with additive noise.  Their experiments reveal that largely biased soft labels can also teach good students and the proposed indicators are effective in measuring the effectiveness of soft labels.

**Strengths:**

* The motivation of the paper is original and novel. It is intriguing to see that learning from largely biased soft labels can achieve comparable performance, and the proposed indicators can be a valuable contribution to measuring their effectiveness in learning a good student for downstream tasks.
* The paper provides an in-depth theoretical guarantee of the biased soft labels in three weakly supervised learning settings.
* The experiment results validate that the proposed indicators (unreliability and ambiguity degree) are indeed effective in measuring the effectiveness of the soft labels (i.e. high accuracy of students) across the three weakly supervised learning paradigms.
* Overall, the paper sufficiently informs the readers of technical and implementation details.

**Weaknesses:**

* The paper shows limited experiment results on CIFAR-10 and CIFAR-100 (Figure 2). The authors should perform experiment on a wider range of benchmark datasets to validate the generality of the method. Are the proposed indicators effective in more complex datasets as the authors suggested?
* The introduction could be better organized. The paper should clearly present the definition of a large-biased soft label and better motivate the readers on why utilizing them is valuable.
* The paper exceeds the nine page limit.

**Questions:**

* Does the proposed indicators work well across other benchmark datasets?

**Limitations:**

The authors did not address the limitations of their work.

---

> ### Author Rebuttal · Authors · 2023-08-08
>
> Dear Reviewer,
>
> Thank you for your constructive feedback. We appreciate the time and effort you took to review our paper, and we value the insights you've provided.
>
> First and foremost, we would like to express our sincere gratitude for your positive remarks on the originality, theoretical depth, and experimental validation of our work. Your recognition of the novelty of our motivation, the in-depth theoretical guarantees provided, and the effectiveness of our proposed indicators is truly encouraging and reinforces our belief in the significance of our contributions. We would like to address the concerns you raised:
>
> **Q1. Experiments on more complex datasets.**
>
> - A: We have supplemented our research with experiments on the Tiny-ImageNet dataset, and the results are illustrated in the PDF of Author Rebuttal.
>
> **Q2. Definition of a large-biased soft labels.**
>
> - A: We apologize for any confusion caused by the term "large-biased soft labels" mentioned in the paper. Here, we provide a rigorous definition for "large-biased soft labels" to clarify its meaning.
>
>   > Definition 1. (Bias of soft labels) Give a dataset $D$ consisting of $n$ samples, the feature vector for the $i$-th sample is denoted as $\boldsymbol{x}_i$ and the corresponding label is denoted as $y_i$. Let $f$ represent a model or a mapping rule (e.g. label smoothing). The bias of the soft labels generated by $f$ on dataset $D$ is
>
>   \begin{aligned}
>   Bias(f, D)=\frac{1}{n} \sum_{i=1}^n [1-f_{y_i} (\boldsymbol{x}_i)],
>   \end{aligned}
>
>   where $f_{y_i} (\boldsymbol{x}_i)$ refers to the component of the soft label $f(\boldsymbol{x}_i)$ that corresponds to the true label $y_i$.
>
>   > Definition 2. (Large-biased soft labels) Soft labels generated by $f$ on dataset $D$ is called biased soft labels when $Bias(f, D) \textgreater 0$ and called large-biased soft labels when $Bias(f, D) \geq 0.5$.
>
> **Q3. The paper exceeds the nine page limit.**
>
> - A: We apologize for the oversight regarding the page limit. We will carefully revise the paper to ensure it adheres to the limit while retaining the essential content and contributions.
>
> Your constructive feedback is instrumental in refining our paper, and we are committed to making the necessary improvements. Once again, thank you for your valuable insights and for considering our paper for acceptance.

---

> > ### Comment · Reviewer_M44N · 2023-08-16
> >
> > Dear authors,
> >
> > Thank you for your detailed response. After reading the response and comments from other reviewers, I have decided to retain my original score.

---

### Official Review · Reviewer_XfEm · 2023-07-11

**Soundness:** 3 good
**Presentation:** 3 good
**Contribution:** 4 excellent
**Rating:** 6
**Confidence:** 5

**Summary:**

This paper explores the concept of learning from weak, or "soft," labels that may be biased or diverge from the ground truth labels in a dataset. In contrast to existing theories that focus on the importance of close alignment between soft and ground truth labels, the authors probe the efficacy of learning from significantly biased soft labels. They introduce two indicators to gauge the effectiveness of these soft labels and propose conditions under which learning from such labels can be successful, including large-biased labels.

The authors further devise a heuristic method for training what they call Skillful but Bad Teachers (SBTs), referring to models with relatively low accuracy that can nevertheless effectively train high-performing student models. They show that these teachers can achieve up to 90% accuracy on the CIFAR-10 dataset, demonstrating the validity of their approach.

Additionally, the authors adapt their theoretical framework to examine the utility of soft labels in three specific weakly-supervised learning paradigms: incomplete supervision, partial label learning, and learning with additive noise. Experimental results are presented to support the proposed indicators and the viability of biased soft labels in these scenarios.

Key contributions include:

    Discovery that learning from largely biased soft labels can achieve comparable performance, and an exploration of the mechanisms behind this phenomenon.
    The proposal of two indicators to evaluate the effectiveness of soft labels and conditions to ensure their usefulness.
    A heuristic method to train SBTs, using new concepts of unreliability degree and ambiguity degree.
    A theoretical framework that illuminates the role of soft labels in three weakly-supervised learning paradigms, accompanied by theoretical guarantees for their learnability and supporting experimental results.

**Strengths:**

Theoretical Analysis: The paper provides a comprehensive theoretical framework for analyzing the effectiveness of soft labels, which are often employed in the realm of machine learning for teaching student models.

Definitions and Indicators: The paper introduces and defines new concepts like unreliability degree and ambiguity degree, and relates them to the effectiveness of soft labels. This could provide valuable insight for the development of future machine learning models.

Extension to Weakly-Supervised Learning (WSL): The research effectively applies the theoretical findings to weakly-supervised learning paradigms, thus demonstrating the applicability and extensibility of their findings.

**Weaknesses:**

Selective Conditions: While the paper provides conditions for classifier-consistency and ERM learnability, it doesn't clearly outline how to meet these conditions in a real-world context. Furthermore, the condition of balancing unreliability degree and ambiguity degree might be challenging to achieve in practice.

**Questions:**

It's mentioned that the soft labels generated by SBTs are large-biased, which could potentially impact the model's ability to generalize to new, unseen data. While the authors note that students still have good accuracy despite the bias, how does the authors propose the current approach tackles this situation?



**Limitations:**

The process of inhibiting correct predictions, reducing unreliability and ambiguity degrees, and randomly selecting k-1 labels for each training instance could increase the computational complexity and time required to train the models.

The evaluation is mainly based on the accuracy of the student models, which might not be the most comprehensive measure. There could be other performance metrics that are important in the context of this problem.

---

> ### Author Rebuttal · Authors · 2023-08-08
>
> Dear Reviewer,
>
> Thank you for your constructive feedback. We appreciate the time and effort you took to review our paper, and we value the insights you've provided.
>
> Your acknowledgment of our paper's strengths, especially the theoretical analysis, the introduction of new indicators, and the extension to weakly-supervised learning, is both encouraging and motivating. It reinforces our belief in the significance of our contributions and provides us with valuable insights for further refinement. We would like to address the concerns and questions you raised:
>
> **Q1. Conditions in Theorem 1 and 2 might be challenging to achieve in practice.**
>
> * In fact, for models that are trained normally, the conditions we described (i.e. $\gamma_k(f)<1-\frac{\eta_k(f)}{1-\eta_k(f)}$ and $\eta_k + \gamma_k<1$) are easily met. It only requires the existence of a specific 'k' such that both delta and gamma are satisfied. To further illustrate this point, when we set $k=4$ and use a normally trained resnet-50 as an example, we obtained the following results:
>   - CIFAR10: Accuracy (ACC): 95.29%; $\eta$: 2.63\%; $\gamma$: 20.12\%.
>   - CIFAR100: Accuracy (ACC): 78.13%; $\eta$: 6.55; $\gamma$: 21.01\%.
>   - Tiny-Imagenet: Accuracy (ACC): 60.63%; $\eta$: 13.55; $\gamma$: 24.67\%.
>
>   They all easily satisfy the requirements stipulated in Theorem 1 and Theorem 2.
>
> **Q2. How does the proposed algorithm make it possible for bad teachers to teach good students?**
>
> * The algorithm proposed in Section 5.1 is inspired by our Theorem 1 and 2. While the soft labels generated by SBTs are large-biased, they exhibit a lower unreliability degree and ambiguity degree, as elaborated in Section 5.1. Theorems 1 and 2 offer assurances for the effectiveness of such large-biased soft labels, thereby enabling the effective training of competent students.
>
> **Q3. The proposed algorithm increase the computational complexity and time in training the models.**
>
> * Based on the logs from our experiments, our method operates within 10% slower than normally trained models. Moreover, the GPU memory consumption is almost identical. Thus, the computational complexity and time required for our approach should be considered acceptable.
>
> **Q4. Evaluation is mainly based on the accuracy of the student models.**
>
> * As illustrated in Table 1 of the Author Rebuttal, we tested four different metrics — Chebyshev distance, KL divergence, Manhattan distance, and Euclidean distance — to measure the discrepancies between the large-biased soft labels generated by SBTs and the ground-truth labels. Furthermore, we also test soft labels generated by a teacher trained under full supervision. What we observed was that, based on these four classical metrics, the soft labels generated by SBTs differ significantly from those of the normally trained teacher. Yet, they still manage to train good students. These classic metrics do not adequately capture the teaching capabilities inherent in soft labels, nor do they reflect the performance of the student models. This finding accentuates our claim that the indicators we introduced are more indicative of the teaching capabilities of soft labels compared to these classic metrics.
>
> Your constructive feedback is instrumental in refining our paper, and we are committed to making the necessary improvements. Once again, thank you for your valuable insights.

---

> ### Comment · Area_Chair_tMq9 · 2023-08-20
>
> Dear Reviewer XfEm,
>
> This is another friendly reminder to acknowledge that you have read the rebuttal and the other reviews. Please also share how they change your view on the paper, if at all. Thanks again for your service!
>
> Best,
>
> AC

---

### Author Rebuttal · Authors · 2023-08-09

We provided three supplementary experiments in the PDF:

* In Figure 1, we show efectiveness of the proposed indicators on Tiny-Imagenet. As $\eta_k(f)$ and $\gamma_k(f)$ decrease, Acc (i.e., accuracy of the student) increases. The experiments on Tiny-Imagenet demonstrate a trend similar to that on CIFAR-10 and CIFAR-100.

* Figure 2 demonstrates whether the effectiveness of soft labels is influenced by different backbones. We experimented with wideresnet 28x2, 28x4, 40x2, and 40x4. In the Figure 2, we omitted the Unreliability degree and Ambiguity degree (which is the same the that of Figure 2 in the paper) for clarity. The four distinct backbones displayed consistent trends, further suggesting that the proposed indicators are effective across different backbones.

* In Table 1, we tested four different metrics — Chebyshev distance, KL divergence, Manhattan distance, and Euclidean distance — to measure the discrepancies between the large-biased soft labels generated by SBTs and the ground-truth labels. Furthermore, we also test soft labels generated by a teacher trained under full supervision. What we observed was that, based on these four classical metrics, the soft labels generated by SBTs differ significantly from those of the normally trained teacher. Yet, they still manage to train good students. These classic metrics do not adequately capture the teaching capabilities inherent in soft labels, nor do they reflect the performance of the student models. This finding accentuates our claim that the indicators we introduced are more indicative of the teaching capabilities of soft labels compared to these classic metrics.

---

### Decision · Program_Chairs · 2023-09-21

**Decision:**

Accept (poster)

**Comment:**

The paper starts with an interesting and original motivation, defines relevant definitions like "unreliability degree" and "ambiguity degree" on the way, and finally provides a comprehensive theoretical framework with applications to weakly-supervised learning. There are certain limitations, such as increased computational complexity and limited validation on CIFAR-10/100, that remain. However, as reviewers have acknowledged, the authors have effectively addressed the most critical concerns. There's no major con against accepting the paper.